# Mini-o3: Scaling Up Reasoning Patterns and Interaction Turns for Visual Search

**Xin Lai**[1*]   **Junyi Li**[1,2*]   **Wei Li**[1]   **Tao Liu**[1]   **Tianjian Li**[1]   **Hengshuang Zhao**[2†]

[1]ByteDance   [2]The University of Hong Kong

## Abstract

Recent advances in large multimodal models have leveraged image-based tools with reinforcement learning to tackle visual problems. However, existing open-source approaches often exhibit monotonous reasoning patterns and allow only a limited number of interaction turns, making them inadequate for difficult tasks that require trial-and-error exploration. In this work, we address this limitation by scaling up tool-based interactions and introducing Mini-o3, a system that executes deep, multi-turn reasoning—spanning tens of steps—and achieves state-of-the-art performance on challenging visual search tasks. Our recipe for reproducing OpenAI o3-style behaviors comprises three key components. First, we construct the Visual Probe Dataset, a collection of thousands of challenging visual search problems designed for exploratory reasoning. Second, we develop an iterative data collection pipeline to obtain cold-start trajectories that exhibit diverse reasoning patterns, including depth-first search, trial-and-error, and goal maintenance. Third, we propose an over-turn masking strategy that prevents penalization of over-turn responses (those that hit the maximum number of turns) during reinforcement learning, thereby balancing training-time efficiency with test-time scalability. Despite training with an upper bound of only six interaction turns, our model generates trajectories that naturally scale to tens of turns at inference time, with accuracy improving as the number of turns increases. Extensive experiments demonstrate that Mini-o3 produces rich reasoning patterns and deep thinking paths, effectively solving challenging visual search problems.

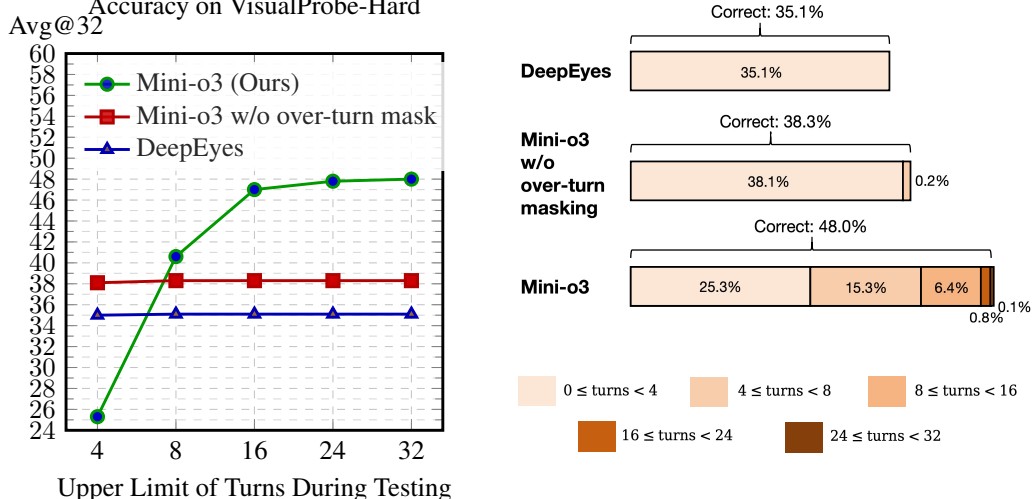

Figure 1: **Left**: Visual search accuracy continues to grow as the upper limit on the number of turns increases for Mini-o3. **Right**: Distribution of the correct trajectories under different numbers of interaction turns during testing. Mini-o3 demonstrates deeper thinking paths and stronger performance. Despite a small upper limit (i.e., 6 turns) during training, it shows the test-time turns scaling property: accuracy continues to grow as the maximum number of turns increases from 4 to 32.

---

[*]Equal Contribution
[†]Corresponding Author

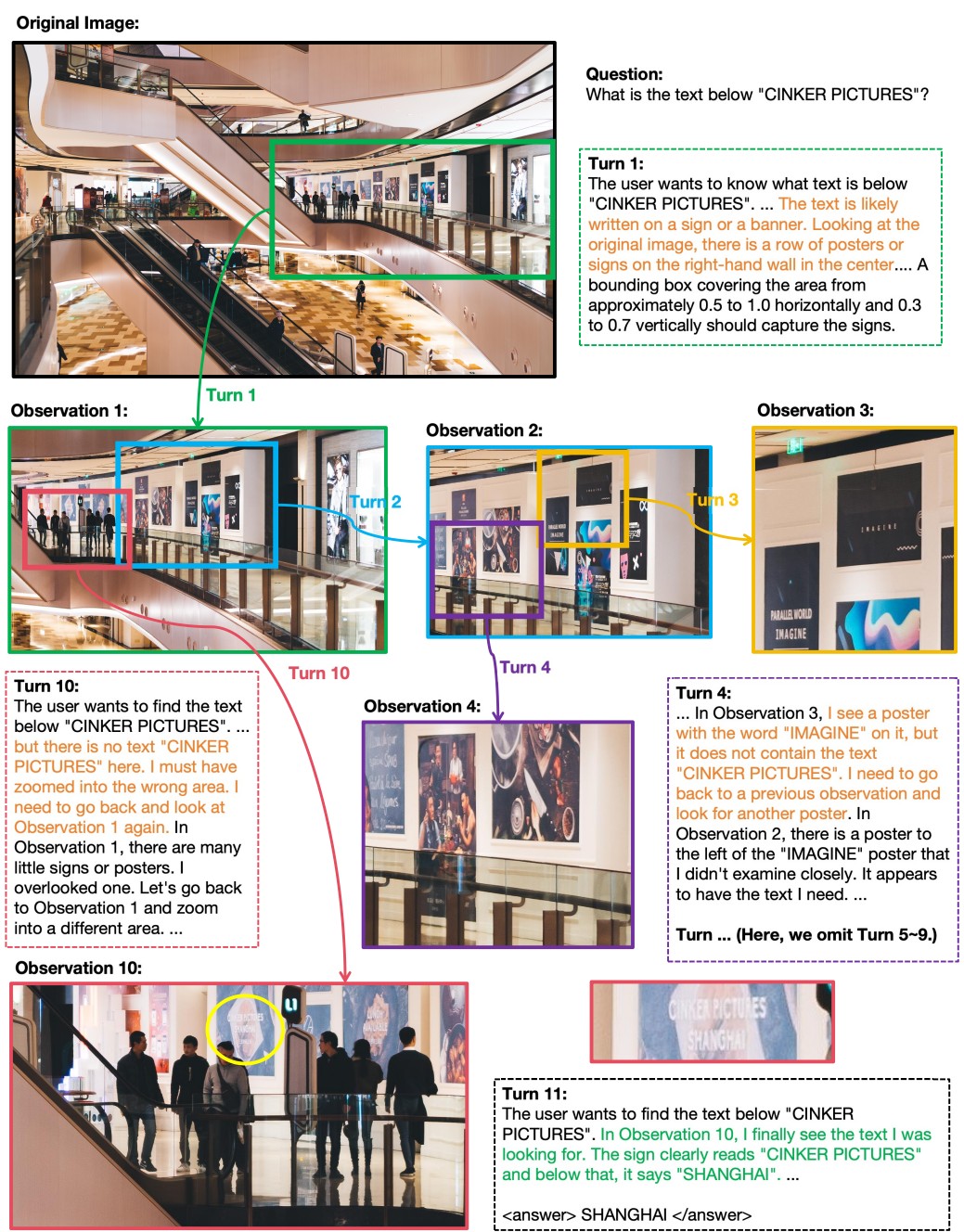

Figure 2: A multi-turn trajectory generated by Mini-o3. It shows complicated reasoning patterns (e.g., trial-and-error exploration) and deep thinking paths (i.e., 11 turns) in visual search tasks. More illustrations are given in Appendix.

# 1 INTRODUCTION

Recently, the capability to invoke image-centric tools has been incorporated into a wide range of Vision–Language Models (VLMs). This thinking-with-image capability enables flexible visual operations and fine-grained reasoning, substantially advancing visual understanding.

However, while existing open-source VLMs exhibit solid performance on relatively simple visual search benchmarks (e.g., V* Bench (Wu & Xie, 2024), HR-Bench (Wang et al., 2025)), they re-

main weak on challenging tasks that require trial-and-error exploration. As shown in Fig.1, Deep-Eyes(Zheng et al., 2025b) achieves only 35.1% accuracy on VisualProbe-Hard. We further observe that this underperformance on difficult problems stems from monotonous reasoning patterns and limited interaction turns. For instance, in HR-Bench-4K, DeepEyes uses image tools for an average of merely one turn per example. Unlike OpenAI o3 (OpenAI, 2025), these models fail to produce diverse reasoning strategies (e.g., depth-first search, trial-and-error exploration, self-reflection) and deep thinking trajectories spanning tens of tool-interaction rounds.

Motivated by these observations, we present Mini-o3 and provide a complete recipe to reproduce the thinking-with-image capability with behaviors similar to OpenAI o3. As illustrated in Fig.2, Mini-o3 generates complex reasoning patterns and deep interaction trajectories, delivering unprecedented performance on challenging visual search tasks. Moreover, Fig.1 (left) demonstrates Mini-o3's ability to scale the number of interaction turns at test time: accuracy consistently improves as the upper bound on interaction turns increases from 4 to 32 during inference, despite training with a budget of only 6 turns. By scaling both the depth of interaction and the diversity of reasoning patterns, Mini-o3 expands the solvable frontier of difficult problems, as shown in Fig. 1 (right).

Our training recipe comprises three components. First, we construct the Visual Probe Dataset, which contains thousands of high-resolution images paired with challenging visual search questions and answers. In contrast to prior benchmarks (e.g., V$^*$ Bench, HR-Bench), where targets are often easy to localize, our problems are explicitly designed to require trial-and-error exploration. Notably, the inclusion of such challenging training samples is essential to elicit diverse reasoning patterns and deep interaction trajectories under reinforcement learning.

Second, we develop an effective pipeline to iteratively synthesize diverse multi-turn trajectories for cold-start supervised finetuning. Concretely, we begin by crafting a small set of representative demonstrations, each comprising the input image and question, along with per-turn observations, thoughts, and actions. These demonstrations cover varied reasoning strategies, including depth-first search, self-reflection, and goal maintenance. We then prompt an existing VLM to mimic these behaviors in a few-shot manner and to produce the thought and action for each turn on new queries, iterating until the model completes the task or reaches the interaction budget. Only trajectories that culminate in a correct answer are retained. Importantly, the base VLM used for data synthesis need not possess native thinking-with-image ability; in-context mimicking suffices.

Third, to enable scaling the number of interaction turns at inference time for harder problems, we avoid penalizing the over-turn trajectories (those that exceed the upper limit of interaction turns) and introduce an over-turn masking technique in reinforcement learning. Specifically, we mask advantages for trajectories that hit the upper limit of interaction turns or the context length. Consequently, over-turn trajectories are ignored during policy updates, and their losses do not contribute gradients. This simple yet effective strategy encourages the emergence of more complex reasoning patterns without overfitting to short trajectories, thereby supporting test-time scaling of interaction depth. It also alleviates the need for a large training-time turn budget: in our experiments, we cap training at only 6 turns, significantly improving efficiency. For example, reducing the training budget from 16 to 6 turns shortens a 10-day training run to about 3 days, with negligible impact on test accuracy.

## 2 RELATED WORK

### 2.1 VISION-LANGUAGE MODELS

The emergence of Vision-Language Models (VLMs) has marked a major milestone in artificial intelligence by enabling the joint understanding of visual and textual modalities. Early seminal works, including BLIP-2 (Li et al., 2023a), Flamingo (Alayrac et al., 2022), and the LLaVA series (Liu et al., 2024; Li et al., 2024a; Guo et al., 2024), established a foundational paradigm that couples strong pre-trained vision encoders (e.g., ViT (Dosovitskiy et al., 2020)) with large language models (LLMs). These systems typically introduce a projector to align visual features with the linguistic embedding space, thereby endowing LLMs with visual grounding. Building on this paradigm, more recent multimodal models—such as Gemini (Team et al., 2023), GPT-4o (Hurst et al., 2024), and Qwen2.5-VL (Bai et al., 2025), among others (Anthropic; Meta; Li et al., 2024b; Chen et al., 2024; Lin et al., 2024)—have achieved state-of-the-art performance on a wide range of visual understanding tasks, notably visual question answering. Their gains are largely driven by scaling model capac-

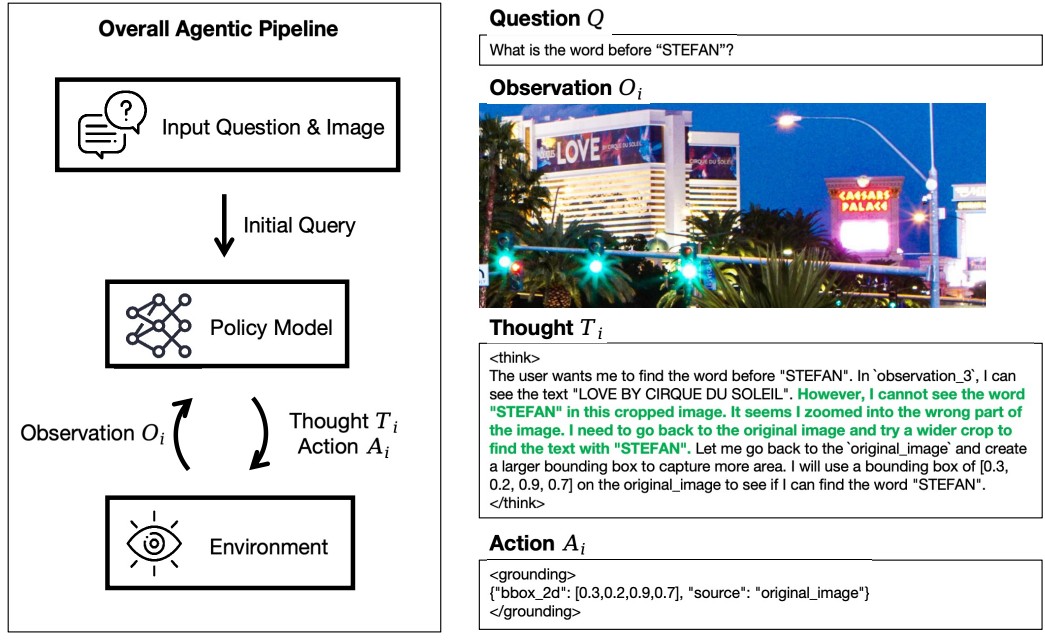

Figure 3: The overview of our framework for multi-turn agentic image tool use. During each turn, the model generates the thought and action iteratively based on the previous observation (or the input question and image). The observation at each turn is obtained based on the parameters indicated by the corresponding action.

ity and training on diverse, high-quality image–text corpora. In parallel, advances in reinforcement learning have enhanced the reasoning capabilities of VLMs by encouraging structured, step-by-step problem solving via Chain-of-Thought prompting (Wei et al., 2022). Recent approaches (Meng et al., 2025; Huang et al., 2025a; Shao et al., 2024a; Liu et al., 2025b; Zhang et al., 2025a; Zhou et al., 2025) primarily target improved textual reasoning for challenging tasks, including counting, logical inference, and mathematical problem solving.

## 2.2  TOOL-INTEGRATED AGENTS WITH REINFORCEMENT LEARNING

Progress in reinforcement learning (RL) including algorithms such as REINFORCE (Williams, 1992), PPO (Schulman et al., 2017), RLOO (Kool et al., 2019), ReMax (Li et al., 2023c), GRPO Shao et al. (2024b), REINFORCE++(Hu, 2025), Dr.GRPO(Liu et al., 2025a), and GSPO (Zheng et al., 2025a) has substantially reshaped training paradigms for both LLMs and VLMs. Systems like DeepSeek-R1 (Guo et al., 2025) and Kimi-K1.5 (Team et al., 2025b) further demonstrated the efficacy of simple, verifiable reward signals in RL for improving reasoning quality. More recently, tool-augmented agents—such as OpenAI's o3 and o4 (OpenAI, 2025), Kimi-Researcher (AI, 2025), Kimi-K2 (Team et al., 2025a), and others (Tao et al., 2025; Geng et al., 2025; Li et al., 2025b; Mai et al., 2025; Xue et al., 2025)—have shown strong agentic abilities in long-horizon, multi-turn tasks by leveraging a broad toolkit (e.g., web browsing, code execution, retrieval). Complementary lines of work, including DeepEyes (Zheng et al., 2025b), Chain-of-Focus (Zhang et al., 2025b), and Pixel Reasoner (Su et al., 2025), as well as related methods (Zhu et al., 2025; Yang et al., 2025; Wu et al., 2025; Huang et al., 2025b), aim to equip VLMs with iterative zoom-in and region-of-interest selection, enabling active perception over images. While these directions collectively point to a promising path for next-generation visual understanding — particularly on challenging, compositional problems — current models often exhibit limited interaction depth and overly rigid reasoning patterns, constraining their effectiveness in complex settings. Our work advances this line by presenting an effective training recipe for a multimodal agent that supports multi-turn image tool use, thereby improving adaptability and reasoning diversity in visually grounded tasks.

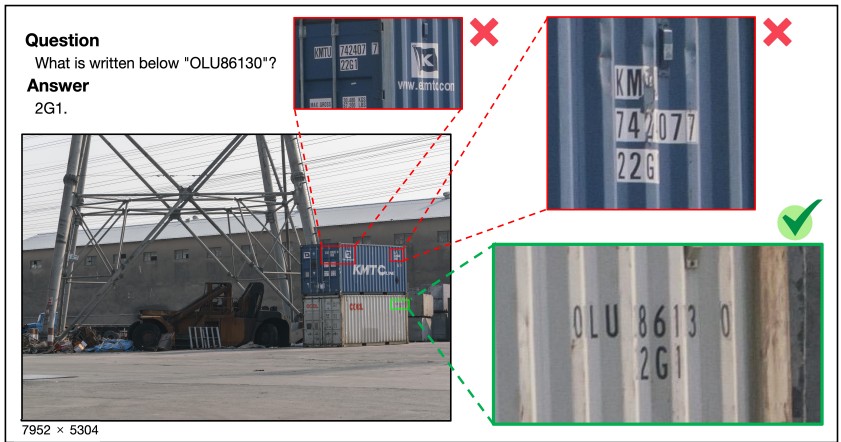

Figure 4: Illustration of the Visual Probe dataset. The Visual Probe dataset features 1) small targets; 2) disturbance objects; 3) high-resolution images. As a result, it is super challenging and requires iterative exploration and trial-and-error.

## 3 OUR APPROACH

### 3.1 OVERVIEW

**Overall Agentic Pipeline** We illustrate the overall agentic pipeline in Fig. 3. Given a user query and an input image, the policy model iteratively produces a thought $T_i$ and an action $A_i$. The action interacts with the environment by invoking image tools, which yields a new observation $O_i$. This observation is appended to the interaction history and fed back to the policy model. The thought–action–observation loop terminates when the model returns a final answer or when predefined limits on context length or interaction turns are reached. The components are detailed below.

- Thought $T_i$: The internal reasoning process used by the policy model to select the next action, conditioned on the interaction history and the current observation. We encourage diverse reasoning patterns within thoughts to facilitate trial-and-error exploration for challenging problems.

- Action $A_i$: The action space comprises two options: (1) grounding and (2) emitting a final answer. For grounding, we parameterize the action with: `bbox_2d`: The normalized bounding box in $[0, 1]^2$ specifying the zoom-in region. `source`: The image on which the grounding operates, chosen from 'original_image" or 'observation_i". This design allows the model to act on any prior observation in the trajectory.

- Observation $O_i$: The observation produced by executing $A_i$ in the environment. Concretely, it is the image patch cropped either from the original image or from a historical observation.

**Two-phase Training** Our training procedure consists of two phases.

- Supervised Fine-Tuning (SFT): We first fine-tune the model on thousands of multi-turn trajectories involving image tool use (i.e., cold-start data). The objective is to teach the model to generate valid trajectories with diverse and robust reasoning patterns.

- Reinforcement Learning with Verifiable Rewards (RLVR): We then apply GRPO (Shao et al., 2024b) to optimize the policy with verifiable, semantics-aware rewards. Because many ground-truth answers in our RL data require semantic rather than exact string matching, we employ an external LLM as a judge to compute reward signals. To maintain training efficiency and stability, we impose upper bounds of 6 interaction turns and a 32K context length.

### 3.2 TRAINING DATA COLLECTION

**Visual Probe Dataset** Hard instances are essential for encouraging reflective, trial-and-error reasoning during reinforcement learning. To this end, we construct a challenging visual search dataset,

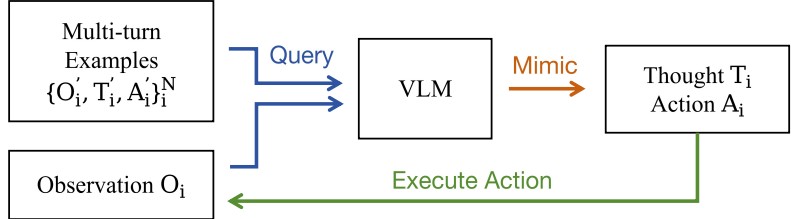

Figure 5: The pipeline of cold-start data collection.

the Visual Probe Dataset (VisualProbe). It comprises $4,000$ visual question–answer pairs for training and $500$ pairs for testing, spanning three difficulty levels: easy, medium, and hard. Compared with prior visual search benchmarks (e.g., V$^*$ Bench), VisualProbe is characterized by: (1) small targets, (2) numerous distractor objects, and (3) high-resolution images, as illustrated in Fig. 4. These properties make the tasks substantially more demanding and naturally require iterative exploration and trial-and-error.

**Diverse Cold-start Data**   We initially attempted to train the model with reinforcement learning alone, without cold-start supervised fine-tuning (SFT). However, the model tended to produce concise responses and trajectories with few turns. We attribute this behavior to the base model's lack of exposure to long-horizon agentic trajectories during pretraining and instruction tuning (here, `Qwen2.5-VL-7B-Instruct`). To handle complex exploratory tasks, we thus employ cold-start SFT to activate multi-turn tool-use capabilities.

The cold-start data collection pipeline is shown in Fig. 5. To generate high-quality, diverse multi-turn trajectories, we prompt an existing VLM with in-context learning ability using a small set of manually crafted exemplars. The VLM is instructed to imitate the exemplars by iteratively producing a thought and an action at each turn. The loop terminates upon emitting a final answer or reaching a pre-defined turn limit. We retain only trajectories whose final answers are correct. Following this procedure, we collect approximately $6,000$ cold-start trajectories from 6 exemplars.

### 3.3   REINFORCEMENT LEARNING

**Lower Down Max Pixels**   The base model's context length is constrained to 32K tokens. With the default image budget of roughly 12M pixels, the allowable number of interaction turns becomes severely limited by context, which hampers trial-and-error exploration on difficult tasks. To increase the feasible turn count per episode, we reduce the maximum pixels per image to 2M (or lower if necessary). This simple adjustment allows more turns to fit within the same context budget, improving solve rates on long-horizon problems.

**Over-turn Masking**   In the vanilla GRPO setting, each question $q$ is passed to the policy model to generate a group of outputs $\{o_i\}_{i=1}^{G}$. Rewards $r$ are then computed based on the correctness of the responses. Notably, when a response hits the maximum number of turns or exceeds the context length limit, the reward is set to $0$, as no valid answer can be produced in such cases. Subsequently, we compute advantages $A$ by normalizing the rewards and update the policy using the GRPO optimization objective over mini-batches. In our implementation, we do not include KL or entropy regularization. Formally, the optimization objective is given by:

$$\mathcal{J}_{GRPO}(\theta) = \mathbb{E}_{[q \sim \mathcal{D}, \{o_i\}_{i=1}^{G} \sim \pi_{\theta_{old}}(\cdot|q)]} \frac{1}{G} \sum_{i=1}^{G} \left( \min \left( \frac{\pi_\theta(o_i|q)}{\pi_{\theta_{old}}(o_i|q)} A_i, \text{clip} \left( \frac{\pi_\theta(o_i|q)}{\pi_{\theta_{old}}(o_i|q)}, 1-\epsilon, 1+\epsilon \right) A_i \right) \right) \tag{1}$$

$$A_i = \frac{r_i - mean(\{r_1, r_2, ..., r_G\})}{std(\{r_1, r_2, ..., r_G\})}. \tag{2}$$

However, we observe that over-turn responses — those that hit the maximum number of turns or exceed the context length — are assigned zero reward, which translates into negative advantages after normalization. In effect, such responses are penalized and discouraged throughout training.

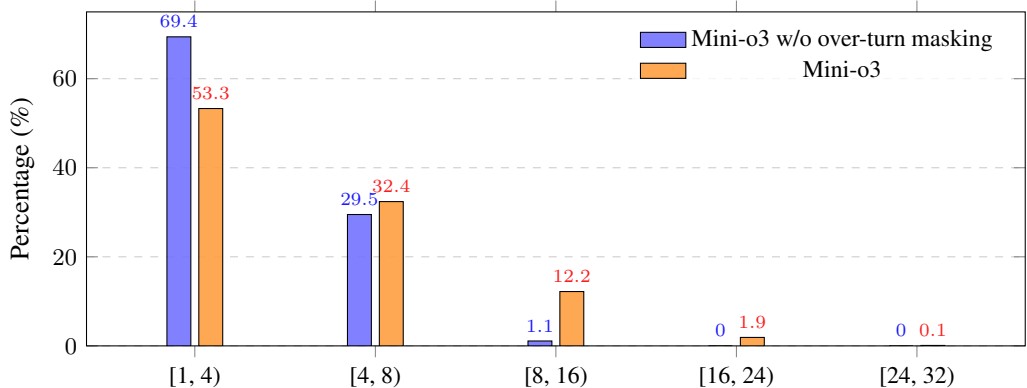

Figure 6: Distribution of interaction-turn percentages across five turn ranges during testing on VisualProbe-Hard. The percentages are calculated only on the *correct* responses.

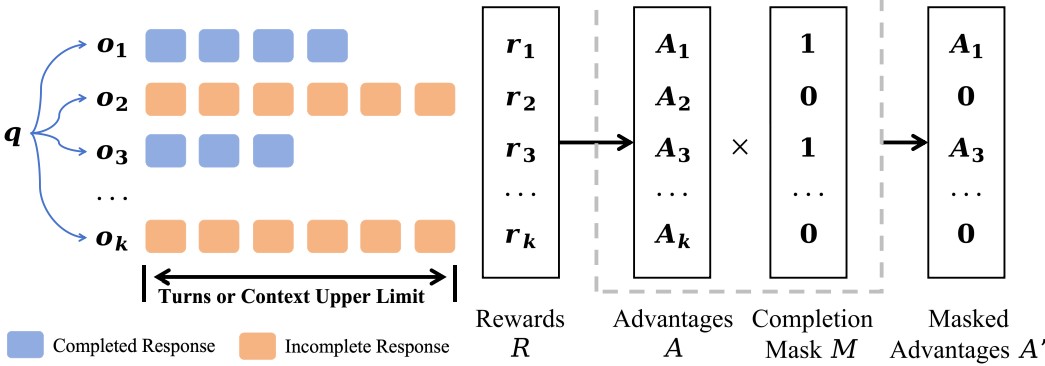

Figure 7: Illustration of the over-turn masking technique. The incomplete responses refer to those that exceed the maximum limit of interaction turns or context length.

This design has two drawbacks. First, the correctness of over-turn responses is inherently unknown; blunt penalization thus injects label noise into the return signal and can destabilize training. Second, for efficiency, the turn limit during training must remain modest (typically fewer than 10 turns). As a consequence, over-turn responses occur frequently — exceeding 20% at the beginning of training. In this regime, naïve penalization biases the model to answer prematurely, substantially suppressing the number of interaction turns (see Fig. 6). This makes highly challenging tasks intractable and severely constrains the potential of test-time scaling.

To prevent the model from collapsing into an "answer earlier" strategy, we propose an over-turn masking technique whose objective is to avoid penalizing over-turn responses. The overall procedure is illustrated in Fig. 7. Concretely, in addition to the rewards $r$ and advantages $A$ defined as in vanilla GRPO, we introduce a completion mask $M$ that indicates whether a response terminates successfully. We then compute masked advantages $A'_i = M_i \cdot A_i$, so that over-turn trajectories (with $M_i = 0$) do not contribute negative learning signals. The modified objective, building on equation 1, is summarized below, with the changes highlighted in red in the formula.

$$\mathcal{J}_{GRPO}^{over-turn}(\theta) = \mathbb{E}_{[q \sim \mathcal{D}, \{o_i\}_{i=1}^{G} \sim \pi_{\theta_{old}}(\cdot|q)]}$$

$$\frac{1}{\sum_i^G M_i} \sum_{i=1}^{G} \left( \min \left( \frac{\pi_\theta(o_i|q)}{\pi_{\theta_{old}}(o_i|q)} A_i \cdot M_i, \mathrm{clip} \left( \frac{\pi_\theta(o_i|q)}{\pi_{\theta_{old}}(o_i|q)}, 1-\epsilon, 1+\epsilon \right) A_i \cdot M_i \right) \right) \tag{3}$$

$$M_i = \mathbb{1}\{|o_i| <= C_{context}\} \cdot \mathbb{1}\{\mathrm{turn}(o_i) <= C_{turn}\}. \tag{4}$$

Here, $|o_i|$ and $\mathrm{turn}(o_i)$ denote the token length and the number of turns in response $o_i$, respectively. Moreover, because some responses are incomplete, we normalize the objective by the number of completed generations, $\sum_i^G M_i$, rather than by the total number of generations $G$.

Table 1: Performance comparisons with existing models on visual search tasks. The sizes of all listed models are 7B. For VisualProbe and V* Bench, we report Avg@32 to reduce variance caused by randomness. We report Avg@8 and Avg@1 for HR-Bench and MME-Realworld, respectively.

| Model | VisualProbe | | | V* | HR-Bench | | MME-Real |
|---|---|---|---|---|---|---|---|
| | hard | medium | easy | | 4K | 8K | |
| GPT-4o (Hurst et al., 2024) | 11.2 | 15.4 | 47.5 | 65.2 | 62.0 | 58.3 | 45.2 |
| LLaVA-OneVision (Li et al., 2024a) | 13.4 | 12.5 | 36.2 | 70.9 | 61.2 | 54.0 | 57.4 |
| Qwen2.5-VL-Instruct (Bai et al., 2025) | 23.9 | 26.0 | 39.1 | 75.5 | 68.2 | 62.7 | 57.3 |
| SEAL[†] (Wu & Xie, 2024) | - | - | - | 75.4 | - | - | - |
| DyFo[†] (Li et al., 2025a) | - | - | - | 81.2 | - | - | - |
| Chain-of-Focus[†] (Zhang et al., 2025b) | - | - | - | 88.0 | - | - | - |
| Pixel Reasoner[‡] (Su et al., 2025) | 28.8 | 29.6 | 58.4 | 86.3 | 74.0 | 66.9 | 64.4 |
| DeepEyes[‡] (Zheng et al., 2025b) | 35.1 | 29.8 | 60.1 | 83.3 | 73.2 | 69.5 | 64.0 |
| Mini-o3 (Ours) | **48.0** | **50.4** | **67.0** | **88.2** | **77.5** | **73.3** | **65.5** |

[†] The models only report the metric of Avg@1 and the model weights are not available.
[‡] Re-evaluated using its official model and evaluation code to yield the metric of Avg@32.

With this technique, we mask out the loss for over-turn responses, thereby removing any implicit penalty. Notably, although we adopt a relatively small upper bound on the number of turns during training, test-time trajectories can extend to dozens of rounds, with accuracy improving monotonically. The proposed over-turn masking is thus essential for realizing the benefits of test-time scaling in the number of interaction turns, as illustrated in Fig. 7.

# 4 EXPERIMENT

## 4.1 EXPERIMENTAL SETTING

**Reinforcement Learning** For reinforcement learning, we follow DAPO (Yu et al., 2025) and adopt clip-higher, dynamic sampling, and a token-level policy loss to ensure stable training. We set the group size to 16. By default, the upper and lower clip ratios are 0.30 and 0.20, respectively. The global batch size is 256, with a mini-batch size of 32. We use a constant learning rate of $1 \times 10^{-6}$. Neither KL regularization nor entropy regularization is applied. To maintain training efficiency, we cap the maximum number of turns at 6 and set the maximum context length to 32K tokens. We also implement asynchronous rollouts to accelerate training.

**Dataset** For training, we use the VisualProbe training split. In addition, to preserve performance on simpler visual search cases, we randomly sample $4,000$ examples from DeepEyes-Datasets-47k (Zheng et al., 2025b). The test suites include VisualProbe-test, V* Bench, HR-Bench, and MME-Realworld (Zhang et al., 2024b).

**Evaluation Metric** We find that single-run evaluation exhibits high variance and does not reliably reflect robustness due to sampling stochasticity. To mitigate this, we report the Avg@K metric: each problem is evaluated $K$ times with temperature set to 1.0, and accuracy is computed by averaging across the $K$ responses.

## 4.2 MAIN RESULT

The performance comparison between existing models and Mini-o3 on visual search tasks is presented in Table 1. To ensure robust and convincing evaluation, we assess all models on VisualProbe, V* Bench, and HR-Bench. Across all datasets, Mini-o3 achieves state-of-the-art performance, substantially outperforming other open-source baselines. We attribute these gains to Mini-o3's ability to sustain more complicated and deeper reasoning trajectories.

## 4.3 ABLATION STUDY

In this section, we present an extensive ablation study to quantify the contribution of each component in our method. The overall results are summarized in Table 2. Unless otherwise specified, all experiments are conducted on the VisualProbe test set with the maximum pixel budget set to 1M.

Table 2: Ablation study for main components of the method. Max pixels are set to 1M. Upper limit on the number of turns is set to 6 during training. Evaluations are made on VisualProbe test set.

| ID | hard RL data | cold-start | over-turn | Hard | Medium | Easy | Avg. Turns (correct) |
|----|--------------|------------|-----------|------|--------|------|----------------------|
| 1  |              | ✓          | ✓         | 35.8 | 46.4   | 66.7 | 4.8                  |
| 2  | ✓            |            | ✓         | 25.4 | 18.7   | 57.3 | 1.0                  |
| 3  | ✓            | ✓          |           | 32.2 | 45.7   | 61.1 | 3.0                  |
| 4  | ✓            | ✓          | ✓         | **44.4** | **47.9** | **67.4** | **5.5**         |

Table 3: Ablation study on the values of max pixels. Evaluations are made on VisualProbe test set. Also, we calculate the average number of interaction turns among overall and correct trajectories.

| Max Pixels | Hard | Medium | Easy | Avg. Turns (All) | Avg. Turns (Correct) |
|------------|------|--------|------|------------------|----------------------|
| 0.5M       | 36.4 | 44.8   | 64.8 | **8.0**          | **6.7**              |
| 1M         | 44.4 | 47.9   | **67.4** | 6.3          | 5.5                  |
| 2M         | **48.0** | **50.4** | 67.0 | 6.5          | 5.6                  |
| 12M        | 36.1 | 40.7   | 62.1 | 1.0              | 1.0                  |

**Hard RL Data**    We compare experiments 1 and 4 in Table 2. Removing the hard RL data leads to a performance decrease of approximately 8.6 points on VisualProbe-Hard, indicating that challenging RL samples are crucial for encouraging complex reasoning trajectories.

**Cold-start SFT**    To assess the necessity of cold-start SFT, we contrast experiments 2 and 4 in Table 2. The results show that cold-start SFT is essential for multi-turn tool use: performance collapses without it. We hypothesize that the base model lacks exposure to multi-turn agentic trajectories during pre-training or instruction tuning, and cold-start SFT serves as a pivotal initialization.

**Over-turn Masking**    A comparison between experiments 3 and 4 in Table 2 demonstrates that over-turn masking benefits reinforcement learning, particularly in multi-turn settings. It offers two main advantages. First, it stabilizes training by avoiding incorrect penalization of truncated responses whose correctness is inherently uncertain. Second, it enables test-time turn scaling and unlocks strong performance on highly challenging tasks that require substantially more turns than the training-time upper bound. This trend is further corroborated in Fig. 6.

**Max Pixels**    Table 3 evaluates different maximum pixel budgets. We observe that both overly large and overly small settings are suboptimal. An excessively large budget induces premature "early stopping", reducing the number of interaction turns and limiting iterative refinement. Conversely, a small budget increases perceptual hallucinations. We also report the average number of interaction turns in the same table, which highlights a trade-off between perceptual accuracy and interaction depth. Optimal overall performance is achieved by appropriately tuning the max-pixel budget.

## 5    CONCLUSION

In this work, we investigate multi-turn image-based tool use for Vision-Language Models (VLMs). To address challenging visual search problems that demand iterative exploration and trial-and-error, we introduce Mini-o3, a model capable of producing diverse reasoning patterns and deep chains of thought. Its trajectories scale to tens of turns, during which accuracy continues to improve, yielding substantial gains over prior models on multiple visual search benchmarks. To enable these capabilities, we develop a three-pronged approach. First, we construct VisualProbe, a challenging visual search dataset comprising both training and evaluation tasks. Second, we devise a simple yet effective pipeline for collecting cold-start data by leveraging the in-context learning ability of an existing VLM. Third, we enhance vanilla GRPO with an over-turn masking strategy that prevents undue penalties on responses that exceed the training budget on turns. This modification facilitates test-time turn scaling and enables the solution of particularly difficult problems. We believe this recipe offers practical guidance for reinforcement learning and the development of multimodal models with multi-turn interactions.

## ACKNOWLEDGEMENT

This work is supported by the National Natural Science Foundation of China (No. 62422606, 62441615) and Hong Kong Research Grant Council General Research Fund (No. 17213925).

## ETHICS STATEMENT

This work adheres to high ethical standards in machine learning and computer vision, ensuring transparency, reproducibility, and fairness throughout all experiments.

## REPRODUCIBILITY STATEMENT

To ensure reproducibility, we provide comprehensive implementation details, including models, datasets, and training setups. All code, datasets, and models will be released.

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

# APPENDIX

## A INFERENCE

**Generation with Temperature**   During inference, we observe that greedy decoding tends to produce *repeated words or sentences*, likely because the effective context grows with the number of turns. To mitigate this issue, a simple yet effective method is to set the temperature to $1.0$, which introduces sufficient randomness to reduce repetition without substantially degrading coherence.

## B EXPERIMENTAL SETUP FOR SUPERVISED FINETUNING

During SFT, we use `Qwen2.5-VL-7B-Instruct` as the base model. Given the context-length constraints in multi-turn agentic interactions, we set the maximum pixel budget to 2M unless otherwise specified. We train on approximately $6,000$ cold-start samples for 3 epochs. The learning rate is set to $1 \times 10^{-5}$, and the global batch size is 32.

## C ABLATION ON UPPER LIMIT ON TURNS DURING TRAINING

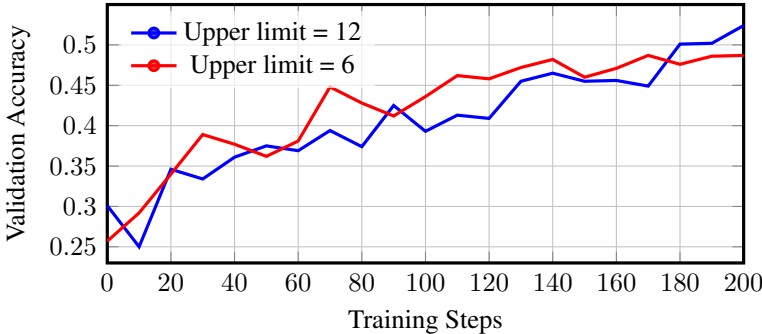

Figure 8: Accuracy on VisualProbe-Hard during the training progress. The upper limit of the number of turns is set to 6 and 12, respectively.

To quantify the effect of a larger interaction-turn budget during training, we track the accuracy on VisualProbe-Hard over the course of training and compare budgets of 6 and 12 turns in Fig. 8. A lower budget leads to faster initial convergence, but the performance plateaus after approximately 150 steps. In contrast, a higher turn budget attains a superior performance ceiling, albeit with slower convergence.

## D PERFORMANCE ON GENERAL VQA BENCHMARKS

To ensure our model maintains strong performance on general Visual Question Answering (VQA) tasks, we incorporate additional VQA data into both the cold-start and reinforcement learning training stages. Our general VQA corpus is filtered primarily from open-source datasets, including LLaVA-OneVision (Li et al., 2024a) and Cambrian-1 (Tong et al., 2024). Note that during the cold-start phase, each VQA sample is paired with a reasoning trajectory generated by Gemini (Team et al., 2023).

We evaluate our model on several general VQA benchmarks, including OCRBench (Liu et al., 2023), ChartQA (Masry et al., 2022), DocVQA (Mathew et al., 2021), MathVista Lu et al. (2023) and POPE (Li et al., 2023b). In addition, we further assess its performance on our VisualProbe-test set, to validate the visual search ability of our model. The detailed results are presented in Table 4. As shown, our model attains state-of-the-art performance on VisualProbe-test while maintaining competitive accuracy across diverse general VQA benchmarks. These outcomes provide clear evidence

Table 4: **Results on General VQA Benchmarks.** We compare the performance of Qwen2.5-VL-7B-Instruct and our model on various general VQA benchmarks, including reasoning, OCR-related tasks and hallucination. We also report their performance on VisualProbe-test. Our model achieves SOTA on VisualProbe-test while maintaining competitive performance on other tasks. Qwen2.5-VL* reports the results evaluated by lmms-eval (Zhang et al., 2024a).

| Category | Benchmark | Qwen2.5-VL* (Bai et al., 2025) | Ours |
|---|---|---|---|
| OCR-Related | OCRBench | 81.5 | 83.8 |
| | ChartQA | 79.6 | 77.4 |
| | DocVQA (val) | 94.6 | 94.8 |
| Reasoning | MathVista (testmini) | 68.2 | 68.8 |
| Hallucination | POPE | 86.7 | 90.8 |
| Visual Search | VisualProbe (easy) | 39.1 | 67.3 |
| | VisualProbe (medium) | 26.0 | 50.5 |
| | VisualProbe (hard) | 23.9 | 48.2 |

that incorporating general VQA data into the training process effectively preserves the model's performance on standard general VQA benchmarks.

# E   DETAILS ON LLM-AS-JUDGE FOR REWARDS.

We use an external LLM as a judge model to evaluate the model's predictions against the ground truth. Notably, we only use pure text for judging during this process. The detailed prompt template is shown below:

Table 5: **Judgment Prompt Template. Question**, **Ground Truth** and **Prediction** are dynamically replaced with the specific question, ground truth and model prediction during evaluation.

---

*SYSTEM PROMPT:*
You are an intelligent chatbot designed for evaluating the correctness of generative outputs for question-answer pairs.
Your task is to compare the predicted answer with the correct answer and determine if they match meaningfully. Here's how you can accomplish the task:
INSTRUCTIONS:
- Focus on the meaningful match between the predicted answer and the correct answer.
- Consider synonyms or paraphrases as valid matches.
- Evaluate the correctness of the prediction compared to the answer.

---

*USER PROMPT:*
I will give you a question related to an image and the following text as inputs:
1. **Question Related to the Image**: **Question**
2. **Ground Truth Answer**: **Ground Truth**
3. **Model Predicted Answer**: **Prediction**
Your task is to evaluate the model's predicted answer against the ground truth answer, based on the context provided by the question related to the image. Consider the following criteria for evaluation:
- **Relevance**: Does the predicted answer directly address the question posed, considering the information provided by the given question?
- **Accuracy**: Compare the predicted answer to the ground truth answer. You need to evaluate from the following two perspectives:
(1) If the ground truth answer is open-ended, consider whether the prediction accurately reflects the information given in the ground truth without introducing factual inaccuracies. If it does, the prediction should be considered correct.
(2) If the ground truth answer is a definitive answer, strictly compare the model's prediction to the actual answer. Pay attention to unit conversions such as length and angle, etc. As long as the results are consistent, the model's prediction should be deemed correct.
**Output Format**:
Your response should include an integer score indicating the correctness of the prediction: 1 for correct and 0 for incorrect. Note that 1 means the model's prediction strictly aligns with the ground truth, while 0 means it does not.
The format should be Score: 0 or 1

---

When evaluating the reward score, we perform **keyword matching** on the evaluation of model's response. Specifically, we first match the pattern "Score: ", and then parse the subsequent number, which is either 0 or 1, to obtain the final reward score.

## F DETAILS ON COLD-START TRAJECTORY GENERATION.

In this section, we present the specific prompt template and corresponding in-context examples that were used in the cold-start data generation phase. We first provide our prompt template as follows:

Table 6: Agent Actions and Generation Format.

| |
|---|
| You are an agent that answers the questions by using following two actions:
1. **grounding([x1, y1, x2, y2], source)**: Return the bounding box [w1, h1, w2, h2] for the region on the 'source' observation image, where (w1, h1) and (w2, h2) are the top-left and bottom-right coordinates (note: w and h represent the width and height; the width and height of the image are normalized to 1.0; the coordinates of top-left corner are (0, 0).), and 'source' could be either "original_image" or "observation_i", meaning one of the history observations;
2. **final(answer)**: return the final answer. |
| Your generation should have the following format:
**Thought**: to analyze the question & observation and inform the action.
**Action**: your action |
| Here are several examples: |

Table 7: Multi-turn Trajectory Example Format.

| |
|---|
| **Example *N***:
**Question**: *Question*
**Original Image**: *Original Image*
**Thought *i***: *Thought*
**Action *i***: *Action*
**Observation *i***: *Observation* |

Table 8: Instructions for Thought Generation.

| |
|---|
| **Thought *i***:
**Action *i***:
Please follow the language style of the above examples to finish the thought and action.
You should mimic human to think carefully and comprehensively. Output the thought **as detailed as possible**, along with some useful description of the observation, suspective conjectures, and underlying world knowledge that might be useful for problem solving.
If the target object is present in observation *i*, please analyze whether final answer is ready or further grounding is needed.
Otherwise, you can go back to either original image or any of the history observations to find another region. In this case, clarify this thinking process, and analyze how the old bounding box deviates from the correct one, and how you obtain the coordinates of the new bounding box in **Thought *i***. |
| The output format should be in JSON as:

```
{
    "thought": "xxx",
    "action": "grounding([x1, y1, x2, y2], source='xxx')"
}
``` |

Below we show the visualization results of the in-context examples used.

**Original Image:**

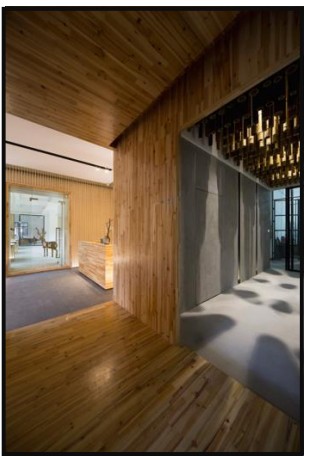

**Question:**
What is written in English on the top first line of the right side of the wooden wall in the middle of the picture?

**Thought 0:**
The user's question asks about the text on the right side of the wooden wall. To analyze it, I may need to zoom into the center part of the image. It seems like the text is too small, so I'll have to crop that region and possibly enlarge the image to read it properly. It could be something like "meeting room" or "conference room," but I'll need to visually inspect and confirm.
I see there might be small metal letters on the wooden wall, likely on the right side. To check the text, I'll crop the image around the dimensions of 0.45-0.65 x 0.3-0.6. After cropping, I'll zoom in to make it easier to read. This should help me see the text clearly, allowing me to provide the most accurate answer based on this zoomed-in section of the image.

**Action 0:**
grounding([0.45,0.3,0.65,0.6], source="original_image")

**Turn1  Observation 1**

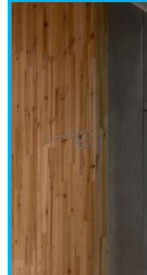

**Thought 1:**
I see small metal letters on the right side of the central wooden wall. To focus on just the text, I'll crop the image more narrowly, focusing only on the text area. By narrowing the crop horizontally, I can zoom in specifically on the text, which should help me read it more clearly. This should allow me to zoom in and get a more detailed and accurate result. The target text is located approximately at the top half of this image, so let's focus on the range from 0.25 to 0.5 of the height dimension in the image of observation 1.

**Action 1:**
grounding([0.05,0.35,0.95,0.55], source="observation_1")

**Turn2  Observation 2**

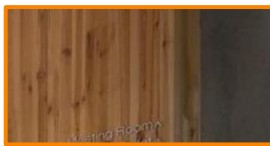

**Thought 2:**
I noticed there might be small metal letters on the right side of the central wooden wall, which is facing the camera. But I could only see the first line of English text partially in observation 2. So I should return to observation 1 and try a better bounding box. The previous bounding box is [0.05,0.25,0.95,0.50] and the resulting observation seems too high, so I should move the top left corner of the bounding box downwards from (0.05, 0.25) to (0.05, 0.35), and the bottom right corner from (0.95, 0.50) to (0.95, 0.55). Therefore, the new bounding box should be [0.05,0.35,0.95,0.55] to better see the text. Once cropped, I can display it and zoom in to get a clearer view of the text. This should help me identify the text in greater detail!

**Action 2:**
grounding([0.05,0.35,0.95,0.55], source="observation_1")

**Turn3  Observation 3**

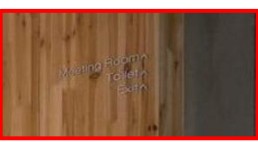

**Thought 3:**
Upon closer inspection, I can clearly see the English text on the top first line is "Meeting Room". The final answer is confirmed. No further checking is needed.

**Action 3:**
final("Meeting Room")

Figure 9: In-context Example 1.

**Original Image:**

**Question:**
What are the black numbers on the yellow license plate?

**Thought 0:**
The user is asking about black numbers on a yellow license plate in an image. Let me locate that plate. It looks like it's a small yellow rectangle on a scooter or cart, on the center-left of the image behind a walking man. I'll focus on that area and zoom in to get a clearer look. I'll display the cropped region, along with a grid for better accuracy, so we can make sure we've found the correct part.
The plate seems to be in the central region, around the left-center, behind a man on a scooter or tricycle. I'll try cropping that central area for a closer look. I'll crop from coordinates around 0.4-0.6 horizontally and 0.4-0.7 vertically. It's a bit tricky, but I'll zoom in and annotate the cropped section before showing it, so I can confirm that I'm targeting the right part of the image.

**Action 0:**
grounding([0.4,0.4,0.6,0.7], source="original_image")

**Turn1   Observation 1**

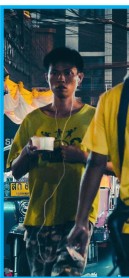

**Thought 1:**
I can't click on the image, so I'll inspect by cropping an approximate area. It looks like the plate is behind a man on a scooter or tricycle, and it's toward the left-center. I focus on cropping that central region (around 0.4-0.6 x 0.4-0.7) and zoom in further to hone in on the license plate, identifying it as a Thai plate with blue and yellow colors. I'll adjust the cropping slightly lower to refine the area. The license plate is approximately at the bottom-left region, around 0.0-0.4 horizontally and 0.6-0.8 vertically. I'll zoom in on this area.

**Action 1:**
grounding([0.0,0.6,0.4,0.8], source="observation_1")

**Turn2   Observation 2**

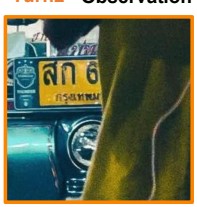

**Thought 2:**
The user wants to know what the black numbers on the yellow license plate are. I can see a "6" clearly. It might be a part of a longer number, but the surrounding characters, possibly in Thai, are obscuring the rest. The license plate likely belongs to a tuk-tuk, as these often have four digits. Thai characters like "กท" seem to represent Bangkok. I'll crop further to check if other digits are visible past the "6" that's currently uncovered. The previous bounding box is `[0.0,0.6,0.4,0.8]`, and we need to extend the bounding box further right. The bottom-right corner of the bounding box is (0.4,0.8), so we should set the width from 0.4 to a larger number, possibly 0.8. Accordingly, the new bounding box should be `[0.0,0.6,0.8,0.8]`.

**Action 2:**
grounding([0.0,0.6,0.8,0.8], source="observation_1")

**Turn3   Observation 3**

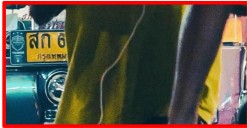

**Thought 3:**
The user asked about the black numbers on the yellow license plate. I can see that the only visible number is "6." There seems to be other Thai characters ("กท") beside it, and possibly more digits to the right, but they're concealed by the man in the image. The only visible black number is "6." I double-checked, and I don't see alternate digits or numbers.
The license plate shows "6" on the right side, possibly followed by more digits, though obstructed. On the left side, there's a gap where the Thai characters "กท" might be visible. Given these observations, it's likely there's more to the plate, but it's being occluded by the man. I'll focus on the visible "6" for answering the question, as it's the only clear number at this moment. The only black digit that's visible on that yellow license plate is a "6." Any other numbers that might be on the plate are blocked by the person standing in front of it.

**Action 3:**
final("6")

Figure 10: In-context Example 2.

**Question:**
How many people are there in total under the two yellow facilities beneath the building on the right?

**Original Image:**

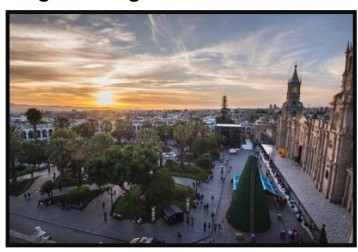

**Thought 0:**
This is an object recognition and counting problem. I need to count the number of people under the two yellow facilities located beneath the building on the right side of the image.
First, I located the building on the right side of the image. It's a building that looks like a church, with a tall tower. Beneath this building, I found two yellow facilities, which look like temporary enclosures or a construction area. I guess the region (0.6-1.0 in width and 0.2-0.8 in height) should include the two yellow facilities. Let's zoom in to see more clearly.

**Action 0:**
grounding([0.6,0.2,1.0,0.8], source="original_image")

**Turn1  Observation 1**

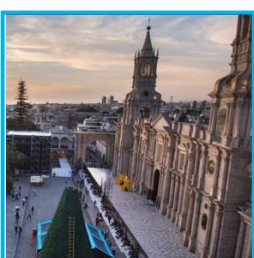

**Thought 1:**
I noticed two yellow facilities beneath the church-like building. Several people are seemingly sitting there and they might be chatting. The user asked how many people there were. There might be three or four people in total, but I couldn't be certain as the region is blurry in this image. To count more accurately, I need to zoom in on the area where these two yellow facilities are located, based on observation 1. They appear in the central region of observation 1, approximately 0.3-0.6 horizontally and 0.5-0.8 vertically.

**Action 1:**
grounding([0.3,0.5,0.6,0.8], source="observation_1")

**Turn2  Observation 2**

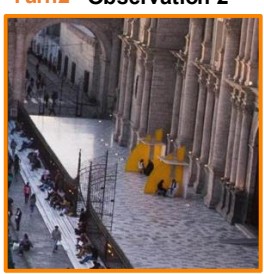

**Thought 2:**
In this further zoomed-in image, I could see the two yellow facilities more clearly, on the porcelain floor. Under each facility, there might be two people talking. Maybe four people in total are seemingly over there. But I couldn't be certain as this region looks blurry up to now. I may need a closer look based on observation 2 to verify. The target area is located at the central of this image, so I'll crop this area (0.4-0.8 horizontally, 0.4-0.8 vertically) and see more details.

**Action 2:**
grounding([0.4,0.4,0.8,0.8], source="observation_2")

**Turn3  Observation 3**

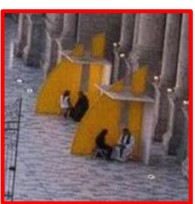

**Thought 3:**
It is clear now. Two people are talking under one yellow facility and another two people are under the other. This verifies that my previous judgement is correct. Now, I can confirm that there are a total of 4 people under the yellow facilities.

**Action 3:**
final("4")

Figure 11: In-context Example 3.

**Question:**
What is the number below the string 207784?

**Original Image:**

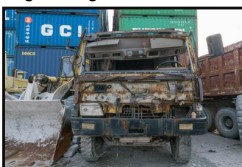

**Thought 0:**
Identify that this is a number recognition task requiring determining the digits in the line below "207784."
Based on my experience, "207784" may be a license plate number. I observe an abandoned truck with its front facing the camera, located in the central part of the image. Maybe "207784" is the license plate number of this truck. I'll crop the front part of this truck to check whether the above hypothesis is correct or not. The front part of the truck is in the center of the image, approximately 0.3-0.8 horizontally and 0.45-0.95 vertically.

**Action 0:**
grounding([0.3,0.45,0.8,0.95], source="original_image")

**Turn1    Observation 1**

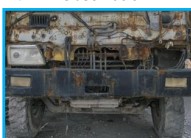

**Thought 1:**
The front part of the truck can be seen in this image. However, there is no license plate number here. Did the bounding box focus on an incorrect region? Wait. let me analyze. I notice the truck lights are present in this crop, including both the left one and the right one. I can confirm that this is indeed the front of the truck. There would be a license plate, but it might be abandoned for a long time and the license plate was lost. Up to now, we confirm that there are no digits "207784" around the abandoned truck. Next, I should continue to search for other areas to locate the target string "207784".
I notice there are numbers in the top-right part of the original image, approximately 0.55-0.85 in width and 0-0.25 in height. Let's zoom in on that area based on the original image.

**Action 1:**
grounding([0.55,0.0,0.85,0.25], source="original_image ")

**Turn2    Observation 2**

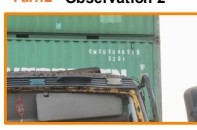

**Thought 2:**
I notice two lines of white characters in this crop, and they are "EMCU3648519" and "22G1". The user asks for the number below "207784". Neither matches the target digit string. So I need to continue searching for the target number.
I observe many white characters in the top-left corner of the original image, which might include the number being sought. Now, let's zoom in on the top-left part (0.0-0.25 horizontally, 0.0-0.5 vertically) of the original image.

**Action 2:**
grounding([0.0,0.0,0.25,0.5], source="original_image")

**Turn3    Observation 3**

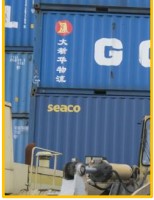

**Thought 3:**
In this crop, I observe many white numbers on the left side, which might include the digits being sought. My target is to find the digits that match "207784". Let's zoom in on the top left corner of observation 3. Based on observation 3, the region is approximately 0.0-0.3 in width and 0.1-04 in height. I'll crop this area with the bounding box `[0.0,0.1,0.3,0.4]`.

**Action 3:**
grounding([0.0,0.1,0.3,0.4], source="observation_3")

**Turn4    Observation 4**

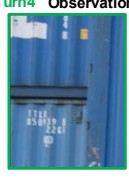

**Thought 4:**
I notice that there are two lines of numbers in the bottom left. They look a little bit blurry, probably "85093922" and "22G1". I am not certain that they are exactly what I said, but one thing to be sure is that neither of them matches the target number "207784".
I need to search other areas to locate the target digits. I notice there is a number "4" at the top of observation 4, which could be part of the number "207784" asked for in the task. I need to look a little upwards to see the entire digit sequence. Let's return to observation 3, and move the previous bounding box a little upwards. The previous bounding box was `[0.0,0.1,0.3,0.4]`. What about moving it upwards by 10% to `[0.0,0.0,0.3,0.3]`?

**Action 4:**
grounding([0.0,0.0,0.3,0.3], source="observation_3")

**Turn5    Observation 5**

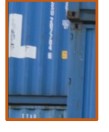

**Thought 5:**
In this crop, I see a vertical line "SU 207784 6" in the top part. The characters seem a little blurry. The user asked for the number "207784". This would be highly likely to be what we are seeking for. The number below looks like "6". I need to double-check by further zooming in on this area. It looks like in the upper part of observation 5, approximately 0.3-0.7 horizontally and 0.0-0.6 vertically. So let's zoom in on this region.

**Action 4:**
grounding([0.3,0.0,0.7,0.6], source="observation_5")

**Turn6    Observation 6**

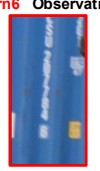

**Thought 5:**
A vertical line reads "SU 207784 6" in this crop, which aligns with my previous judgement, even though it's a little blurry. I can confirm that the number sequence is "207784", and the number directly below it is "6".

**Action 6:**
final("6")

Figure 12: In-context Example 4.

## G   STATEMENT OF LLM USAGE

In this work, an LLM is used to fix grammar errors and polish language expression. But it is not used in technical ideation.

## H   MORE ILLUSTRATIONS OF MULTI-TURN TRAJECTORIES

In this section, we present more examples of the trajectories produced by our model as follows.

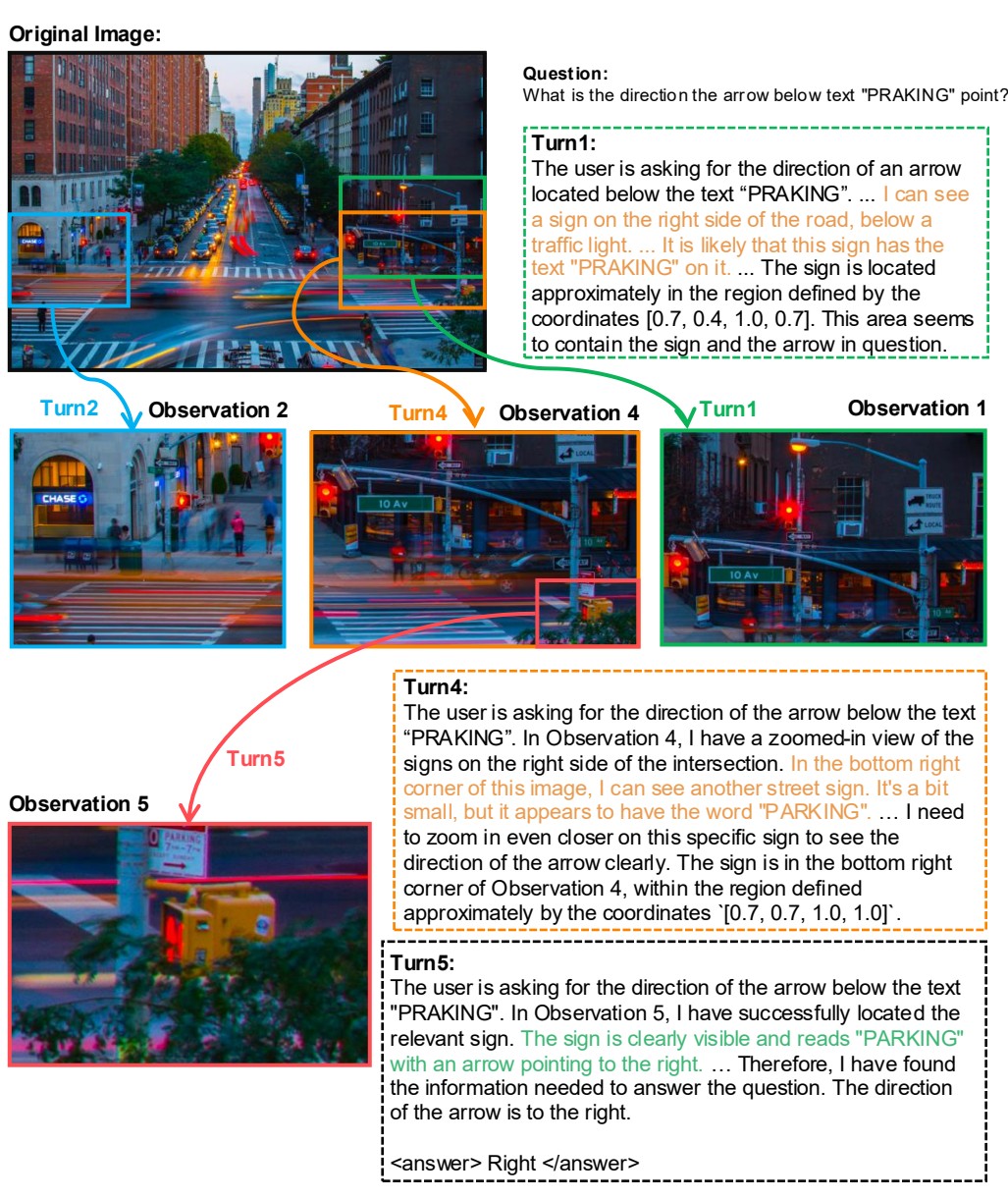

Figure 13: An example of visual search in a urban intersection generated by Mini-o3. The scene shows a busy city street with multiple signs and traffic elements. Our model conducts multi-turn reasoning—progressive zoom-in, hypothesis revision, and backtracking—to identify the direction the arrow is pointing.

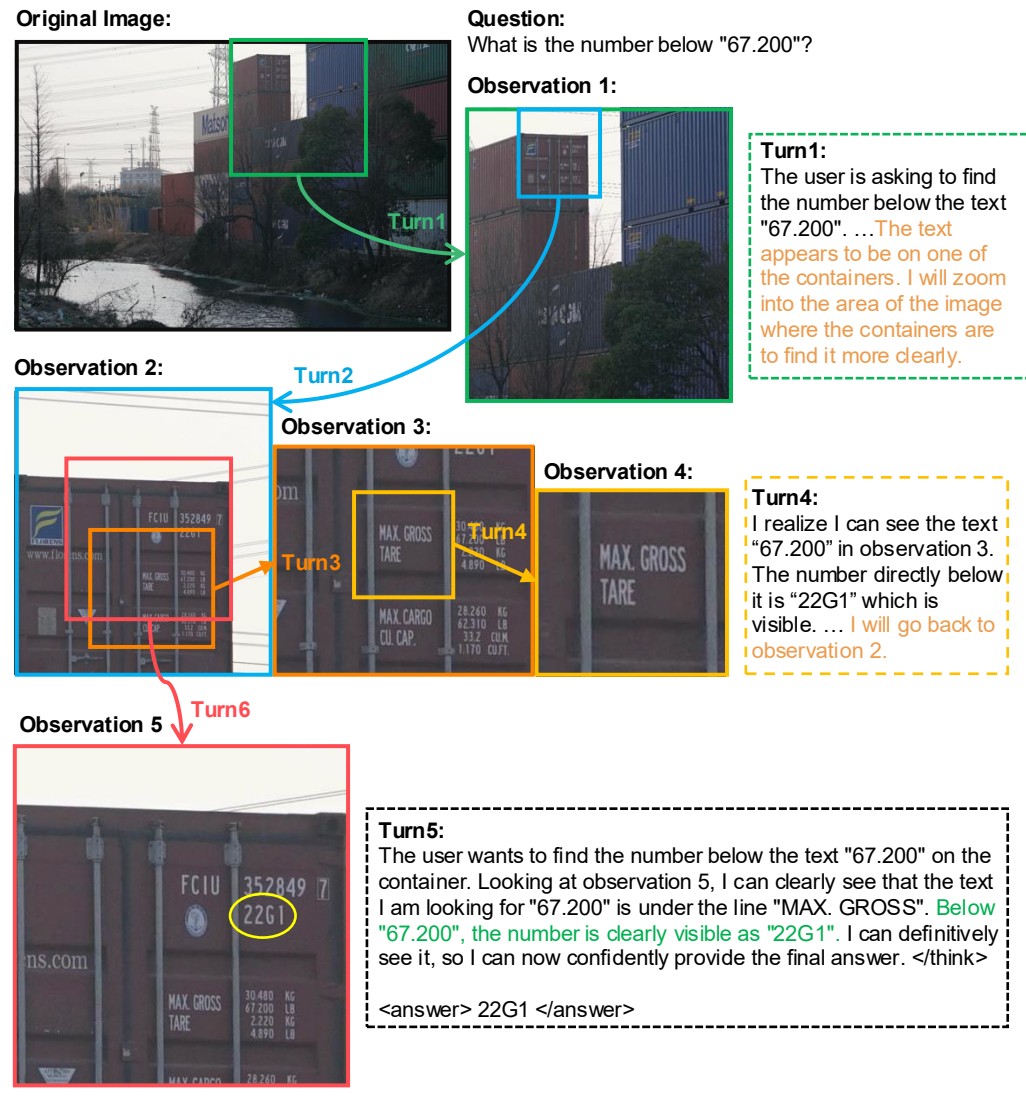

Figure 14: An example of visual search in a container yard generated by Mini-o3. The scene depicts stacked shipping containers with dense text markings. Our model performs multi-turn reasoning—targeted zoom-ins, cross-checking across observations, and corrective backtracking—to locate the string "67.200" and read the number directly beneath it. It outputs the correct value, "22G1", demonstrating precise text localization and robust step-by-step verification in a cluttered setting.

**Question:**
What are the numbers on the circular road sign above the triangular road sign?

Figure 15: An example of visual search in a lakeside village generated by Mini-o3. Our model performs multi-turn reasoning—coarse-to-fine zooming, refocusing, and verification across observations—to localize a circular road sign above a triangular warning sign. Mini-o3 ultimately recognizes the digits "30" on the sign after 18 reasoning turns.

## I    MORE ILLUSTRATIONS OF FAILURE PATTERNS.

In this section, we present examples of two primary failure cases: **Grounding Error** and **Inability to Backtrack**.

**Original Image:**

**Question:**
What color is the hat worn by the child in the **yellow shirt**?
**Ground Truth:** White.

**Ground Truth Region**

**Turn1:**
The user wants to find the color of the hat worn by a child in a yellow shirt. First, I need to locate the area with people in the image to find the child wearing yellow. The main street with people seems to be in the center of the image, so I will crop this area to investigate further.

**Turn1**    **Observation 1**    **Turn2**    **Observation 2**

**Turn2:**
In observation 2, I can identify a child wearing a yellow shirt sitting on the wall. The child appears to be wearing a pink or light-colored hat. To confirm this and determine the exact color with more clarity, I will zoom into the group of children sitting on the wall in the current observation.

**Turn8**    **Observation 8**

**Turn8:**
I can clearly identify the child in `observation 8` and see they are wearing a blue hat. I can confidently say the color of the hat is blue. I can see enough detail to make a final decision, so I can provide the final answer.
<answer> Blue </answer>

**Grounding Error:**
The image shows a child in a white shirt, not a yellow shirt. The model has made a wrong conclusion!

Figure 16: Failure case: **Grounding Error**.

**Original Image:**

**Question:**
What color are the shorts on the person in a **black vest**?
**Ground Truth:** White.

**Ground Truth Region**

**Turn1:**
The user is asking about the color of the shorts of a person wearing a black vest. First, I will locate the person in a black vest in the image. I can spot a person in a black vest near the right side of the image in the middle ground, walking or running. To better see their appearance, I will crop this area.

**Turn1**  **Observation 1**    **Turn2**  **Observation 2**    **Turn3**  **Observation 3**    **Turn4**  **Observation 4**

**Turn4:**
The user wants to identify the color of the shorts of the person wearing the black vest. I have a clear view in observation 4. I can now clearly make out the shorts, and they appear to be blue with some white vertical and horizontal stripes, which suggests a patterned light blue, possibly denim.
<answer> Light blue. </answer>

**Inability to Backtrack:**
The model gets stuck on a single region, neglecting to verify if it's correct.

Figure 17: Failure case: **Inability to Backtrack**.