# OpenReview forum: "Mini-o3: Scaling Up Reasoning Patterns and Interaction Turns for Visual Search"
_ICLR.cc/2026/Conference — ICLR 2026 Poster_

### Official Review · Reviewer_XKVc · 2025-10-29

**Soundness:** 3
**Presentation:** 3
**Contribution:** 3
**Rating:** 6
**Confidence:** 3

**Summary:**

This paper introduces Mini-03, a vision–language model designed for deep, multi-turn visual reasoning in complex visual search tasks. The authors propose a three-part training recipe:

- VisualProbe Dataset – A challenging benchmark with high-resolution images, small targets, and distractors to necessitate iterative reasoning.

- Cold-start Data Pipeline – An iterative synthesis process generating ~6,000 multi-turn trajectories from a few exemplars for supervised fine-tuning.

- Over-turn Masking – A reinforcement learning modification to GRPO that prevents penalization of trajectories exceeding the turn limit, enabling test-time scaling.

Built on Qwen2.5-VL-7B-Instruct, Mini-03 demonstrates emergent test-time turn scaling: trained with 6 turns, accuracy improves up to 32 turns at inference. It achieves state-of-the-art performance on VisualProbe-Hard (48.0% vs. 35.1% for DeepEyes) and competitive results on V*Bench and HR-Bench. Ablations confirm the necessity of cold-start SFT and over-turn masking.

**Strengths:**

- A reproducible recipe combining dataset, data generation pipeline, and RL.

- Over-turn masking is simple yet effective, unlocking test-time scaling—a property with implications beyond vision tasks.

- Promising results across benchmarks, systematic ablations, and insightful analysis of resolution vs. interaction depth trade-offs.

- VisualProbe fills a gap for evaluating exploratory reasoning.

**Weaknesses:**

- The paper does not explain why over-turn masking enables scaling—mode collapse prevention or exploration diversity?

- Relies on a teacher VLM for trajectory synthesis; sensitivity to exemplar quality is unexplored.

- Reward computation details (model choice, prompt, bias analysis) are missing -- reproducibility concerns.

- Focused on visual search; generalization to other reasoning domains (math, scientific figures) is untested – narrow scope.

- Systematic study of error patterns or robustness to adversarial perturbations would strengthen the paper.

**Questions:**

- How sensitive is performance to the choice and diversity of cold-start exemplars?

- Which LLM was used as the reward judge, and how was its reliability validated?

- Can you provide theoretical or empirical analysis of why over-turn masking enables test-time scaling?

- Does the approach generalize to other reasoning domains beyond visual search?

- What are the failure modes—e.g., repetitive loops, grounding errors, or inability to backtrack?

---

> ### Author Response · Authors · 2025-11-22
> **Rebuttal by Authors (Part 1 of 3)**
>
> Dear Reviewer XKVc,
>
> Thank you for your thorough review and insightful feedback. We are grateful for your praise "**reproducible recipe**", "**over-turn masking strategy is simple yet effective**", and "**VisualProbe dataset fills an important gap**". We have revised the paper following your suggestions, and address your questions below.
>
> > ### [Weakness 1 / Question 3] Why Over-turn Masking Enables Scaling？
>
> As we detailed in Lines 322-368 of the submission file, the key reason is that standard reinforcement learning methods (e.g., GRPO) penalize trajectories that exceed the turn limit by assigning them zero reward (which becomes a negative advantage after normalization). This naive penalization forces the model to answer prematurely to avoid punishment, effectively suppressing its reasoning depth. Over-turn masking directly addresses this by masking out the negative learning signals for these incomplete trajectories.
>
> We agree with your hypothesis that this mechanism helps with both mode collapse prevention and exploration diversity.
> - **Mode Collapse Prevention:** By avoiding penalizing long trajectories, our method prevents the model from collapsing into a single, suboptimal "answer earlier" strategy. This allows the model to maintain and develop deeper reasoning pathways.
> - **Exploration Diversity:** Since the model is no longer discouraged from attempting deeper reasoning paths, it is free to explore more diverse and complex strategies during training. This increased diversity in exploration is crucial for discovering the multi-step solutions required for challenging tasks, which in turn enables the model to scale its reasoning depth at test time.
>
> > ### [Weakness 2 / Question 1] Reliance on a Teacher VLM and Sensitivity to Exemplar Quality
>
> We would like to clarify two key aspects: the role of the teacher VLM and the selection of exemplars.
>
> First, regarding the teacher VLM, our method only leverages its **in-context learning ability** to generate cold-start data. As highlighted in Line 135-137, the base VLM used for data synthesis need
> not possess native thinking-with-image ability; in-context mimicking suffices. There are no other specific constraints on the choice of the teacher VLM, meaning other powerful VLMs could also serve this role. The primary goal is simply to bootstrap the initial data generation process.
>
> Second, only 4 exemplars were used in cold-start data generation, and they were selected randomly. We did not manually cherry-pick exemplars. We also tried another randomly selected 4 exemplars to produce the same amount of cold-start trajectories, and yielded similar result after SFT. Therefore, the final performance is not sensitive to the specific choice of exemplars, as long as they are drawn randomly. This also demonstrates the robustness of our data synthesis approach.

---

> ### Author Response · Authors · 2025-11-22
> **Rebuttal by Authors (Part 2 of 3)**
>
> > ### [Weakness 3 / Question 2] Details on Reward Process
>
> Thank you for pointing out the issue of missing details. We list the details below and add them to the appendix of our revised paper.
>
> - **Model Choice:** We use Qwen2.5-VL-72B-Instruct as our external LLM judge for reward computation.
>
> - **Prompt Template:** The system prompt used to guide the LLM judge is provided below. This prompt instructs the model to act as an evaluator and score the correctness of model's prediction against the ground truth.
>
> > **Prompt Template for LLM-as-Judge.** *Question*, *Ground Truth*, and *Prediction* are dynamically replaced with the specific question, ground truth, and model prediction during evaluation.
> >
> > ---
> >
> > **SYSTEM PROMPT:**
> >
> > You are an intelligent chatbot designed for evaluating the correctness of generative outputs for question-answer pairs. Your task is to compare the predicted answer with the correct answer and determine if they match meaningfully. Here's how you can accomplish the task:
> >
> > ---
> >
> > ### INSTRUCTIONS:
> >
> > - Focus on the meaningful match between the predicted answer and the correct answer.
> > - Consider synonyms or paraphrases as valid matches.
> > - Evaluate the correctness of the prediction compared to the answer.
> >
> > ---
> >
> > **USER PROMPT:**
> >
> > I will give you a question related to an image and the following text as inputs:
> >
> > 1.  **Question Related to the Image**: *Question*
> > 2.  **Ground Truth Answer**: *Ground Truth*
> > 3.  **Model Predicted Answer**: *Prediction*
> >
> > Your task is to evaluate the model's predicted answer against the ground truth answer, based on the context provided by the question related to the image. Consider the following criteria for evaluation:
> >
> > - **Relevance**: Does the predicted answer directly address the question posed, considering the information provided by the given question?
> > - **Accuracy**: Compare the predicted answer to the ground truth answer. You need to evaluate from the following two perspectives:
> >   (1) If the ground truth answer is open-ended, consider whether the prediction accurately reflects the information given in the ground truth without introducing factual inaccuracies. If it does, the prediction should be considered correct.
> >   (2) If the ground truth answer is a definitive answer, strictly compare the model's prediction to the actual answer. Pay attention to unit conversions such as length and angle, etc. As long as the results are consistent, the model's prediction should be deemed correct.
> >
> > **Output Format**:
> > Your response should include an integer score indicating the correctness of the prediction: 1 for correct and 0 for incorrect. Note that 1 means the model's prediction strictly aligns with the ground truth, while 0 means it does not. The format should be "Score: 0 or 1".
>
> - **Bias Analysis and Reliability:** Our evaluation process is designed to be a straightforward textual comparison. The LLM judge is only provided with the model's final prediction and the ground truth, both of which are in text format. This simple and constrained setup guarantees an objective and accurate assessment.
>
> To further validate this, we conducted a consistency analysis on our VisualProbe benchmark by comparing the LLM judge's scores against evaluations from human experts. The results showed a **98% agreement rate**, which confirms that the rewards provided by our LLM judge are highly reliable and trustworthy.

---

> ### Author Response · Authors · 2025-11-22
> **Rebuttal by Authors (Part 3 of 3)**
>
> > ### [Weakness 4 / Question 4] Generalization to Other Reasoning Domains
>
> As detailed in Appendix A.4 of the submission file, we have evaluated Mini-o3's performance on several general VQA benchmarks, including MathVista (a chanllenging mathematical reasoning benchmark). Following your valuable suggestion, we conduct additional experiments on scientific figures. The results are shown as follows.
>
> | Model | MathVista | ScienceQA | SciBench |
> | :--- | :--- | :--- | :--- |
> | Qwen2.5-VL-7B-Instruct | 68.2 | 69.4 | 22.5 |
> | Mini-o3 | **68.8** | **84.5** | **23.1** |
>
> As shown in the table, Mini-o3 demonstrates performance gains across these reasoning-focused domains, particularly on the ScienceQA benchmark.
>
> Moreover, as validated in Appendix A.4 of the submission file, incorporating diverse data during training can effectively enhance performance in other domains. Based on this finding, we could enhance generalization on even more domains by scaling data diversity and quantity.
>
> > ### [Weakness 5 / Question 5] Analysis of Error Patterns and Robustness to Adversarial Perturbation
>
> **1. Error Pattern Analysis:**
>
> We analyzed the failure cases and identified two primary error patterns:
> - **Grounding Errors**: Instances where the model fails to accurately link a textual description to the correct visual region in the image.
> - **Inability to Backtrack**: Cases where the model fixates on a single region or gets stuck in a repetitive reasoning loop without verifying its correctness.
>
> To provide a clear view of these behaviors, we have included more visualizations of these failure patterns in our **revised Appendix**. We kindly invite the reviewer to examine these examples.
>
> **2.Robustness to Adversarial Perturbation:**
>
> To evaluate the model's robustness, we tested its performance on images perturbed with **varying levels of Gaussian noise**. We evaluated our standard Mini-o3-7B model on test images perturbed with Gaussian noise at various intensity ratios (5%, 10%, and 20%) without any fine-tuning. The results are summarized in the table below:
>
> | Noise Ratio | VisualProbe-Easy | VisualProbe-Medium | VisualProbe-Hard | V* Bench |
> | :--- | :--- | :--- | :--- | :--- |
> | 0% | **67.0** | **50.4** | **48.0** | **88.2** |
> | 5% | 66.9 | 50.2 | 46.0 | 87.4 |
> | 10% | 66.4 | 49.7 | 43.2 | 84.2 |
> | 20% | 65.6 | 48.5 | 39.8 | 77.9 |
>
> As the results indicate, our model demonstrates strong resilience to noise. With up to 10% Gaussian noise, the model maintains a relatively robust performance across the benchmarks.

---

> > ### Comment · Reviewer_XKVc · 2025-11-25
> >
> > I thank the authors for the responses. I have no further questions. However would be keen to understand author's perspective on why does MathVista and SciBench show only +0.6% gain while ScienceQA shows +15.1%? What properties enable transfer?

---

> ### Author Response · Authors · 2025-11-26
>
> Thanks a lot for your time and efforts. As for the follow-up question, we believe the performance difference primarily stems from difficulty levels and knowledge requirements of the benchmarks.
>
> Since ScienceQA is relatively easy and straightforward (the majority of its questions sourced from the grade 3-8 curriculum), the data distribution is already included in our training data. As a result, the core reasoning skills enhanced by our method are sufficient to unlock a significant performance leap (+15.1\%).
>
> In contrast, SciBench and MathVista contain complex, college-level problems, and deep, domain-specific knowledge is also required. Since our training does not involve large-scale domain-specialized data, the gains are naturally more modest (+0.6\%).

---

### Official Review · Reviewer_iPEo · 2025-10-30

**Soundness:** 3
**Presentation:** 3
**Contribution:** 3
**Rating:** 8
**Confidence:** 3

**Summary:**

The authors propose Mini-o3, a method capable of performing deep, multi-turn reasoning and achieving state-of-the-art performance on challenging visual search tasks.
They constructed a multi-turn visual reasoning trajectory dataset and introduced an over-turn masking strategy to balance training-time efficiency with test-time scalability.

**Strengths:**

The main innovation lies in the proposal of an over-turn strategy. In addition, it demonstrate that although only up to 6-round data  are used during training, the accuracy consistently improves as the upper bound on interaction turns increases from 4 to 32 during inference.

**Weaknesses:**

The setting of the 6-round budget has not been proven to be optimal.

**Questions:**

Have you tried testing the performance of different sizes of QWEN-VL-2.5?

---

> ### Author Response · Authors · 2025-11-22
> **Rebuttal by Authors**
>
> Dear Reviewer iPEo,
>
> We appreciate your valuable feedback and positive evaluation. We are pleased that you acknowledged our Mini-o3's strong performance, particularly its ability to achieve **consistent accuracy improvements with more inference turns**, which demonstrates its **test-time scalability**. We will address your specific questions in the following responses.
>
> > ### [Weakness 1] Optimal Choice of Turn Budget in Over-turn Masking
>
> We agree that a 6-turn budget may not be the absolute optimal setting for achieving the highest possible performance. As we discussed in **Appendix A.3 (Fig. 8)**, our choice of the turn budget reflects a deliberate **trade-off between training efficiency and the final performance ceiling**.
>
> Specifically, while a higher budget (e.g., 16 turns) can lead to a superior performance limit, it comes at the cost of increased computational overhead and slower convergence. As shown in Line 145-146 of the submission file, a 16-turn budget requires 10-day training time, and the training efficiency is unacceptable. However, a 6-turn budget turns out to obtain a sufficiently strong performance at a cost of 3-day training, allowing a practical balance between efficiency and performance.
>
> > ### [Question 1] Performance of Different Base Model Sizes
>
> To investigate the scalability on model size, we apply the proposed training recipe on different sizes of the Qwen2.5-VL model, and the results are shown as follows. All models are trained with the exact same cold-start and reinforcement learning data to ensure a fair comparison.
>
> | Base Model | VisualProbe-Easy | VisualProbe-Medium | VisualProbe-hard |
> | :--- | :--- | :--- | :--- |
> | Mini-o3-Qwen2.5-VL-3B | 57.4 | 40.0 | 42.8 |
> | Mini-o3-Qwen2.5-VL-7B | 67.0 | 50.4 | 48.0 |
> | Mini-o3-Qwen2.5-VL-32B | 71.4 | 57.3 | 51.6 |
>
> As shown in the table, **the model's performance consistently improves with a larger base model.**. This clear upward trend demonstrates the effectiveness of our Mini-o3 framework. It confirms that our approach is not limited to a specific model size but is a robust method that can successfully leverage the enhanced capabilities of larger, more powerful foundation models.

---

### Official Review · Reviewer_ku2Y · 2025-10-30

**Soundness:** 3
**Presentation:** 2
**Contribution:** 3
**Rating:** 6
**Confidence:** 3

**Summary:**

This paper presents **Mini-o3**, an open-source vision-language agent designed to perform **deep multi-turn reasoning** for visual search tasks. While prior open-source VLMs (e.g., DeepEyes, Chain-of-Focus) often exhibit shallow, repetitive reasoning with limited tool-use turns, Mini-o3 demonstrates reasoning trajectories spanning tens of steps. The authors propose a three-component recipe:

1. **VisualProbe dataset**—a new benchmark of high-resolution, hard visual-search problems requiring trial-and-error exploration;
2. **Iterative cold-start data synthesis**—an in-context prompting method for generating diverse reasoning trajectories for supervised fine-tuning;
3. **Over-turn masking**—a reinforcement-learning modification that avoids penalizing long, incomplete trajectories, enabling scalability beyond the training turn limit.

Trained with a 6-turn cap, Mini-o3 generalizes to much longer inference trajectories (up to 32 turns), achieving **state-of-the-art accuracy** on VisualProbe, V* Bench, HR-Bench, and MME-Realworld benchmarks. The model shows more diverse reasoning strategies and deeper “thinking-with-images” patterns than prior open-source agents.

**Strengths:**

* Addresses a **clear and under-explored gap**: enabling deep visual reasoning in open-source VLMs.
* **Elegant and reproducible training recipe** with quantitative and qualitative validation.
* **Comprehensive experiments** (four benchmarks + ablations).
* Demonstrates **turn-scaling property**—a rare capability among existing agents.
* Good ethical and reproducibility practice; code/data promised for release.

**Weaknesses:**

- [Reward Modeling] The paper employs an external LLM as a semantic judge for reinforcement learning but provides few details about its consistency, inter-run variance, or bias. It is unclear how reward noise affects policy stability or whether calibration against human-verified scores was attempted.
- [Ablation] While ablations exist, they focus on quantitative accuracy. The paper could include deeper analyses of emergent reasoning behaviors—e.g., trajectory diversity, failure patterns, or qualitative diagnostics showing how reasoning depth relates to success rates.

**Questions:**

Please see weaknesses above.

1. How sensitive is over-turn masking to the masking threshold or turn budget?
2. Do long-turn trajectories ever degenerate into loops or repetitive zooming?
3. How well would Mini-o3 scale with a larger base model?

Minor Grammar & Typo:

Line 014 - 015: “We address this limitation by scaling up tool-based interactions and introduce Mini-o3…” should be "introducing”

Line 018: “OpenAI o3–style behaviors” should be a regular hyphen

---

> ### Author Response · Authors · 2025-11-22
> **Rebuttal by Authors (Part 1 of 3)**
>
> Dear Reviewer ku2Y,
>
> Thanks for your constructive and insightful feedback. We are grateful that you find our approach to be an **elegant and reproducible recipe for deep visual reasoning** and that **our experiments were comprehensive**, achieving **state-of-the-art results**. We have revised the paper following your suggestions, and address your questions below.
>
> > ### [Weakness 1] Reliability and Consistency of the LLM Judge
>
> Thanks for the insightful question regarding our reward modeling. We would like to clarify the reward process to address your concerns about consistency and reliability.
>
> **During evaluation, the LLM judge is provided with only the final predicted answer and the ground truth, both of which are in text format.** This simplifies the task for the LLM to a straightforward textual comparison, significantly minimizing potential noise and bias, thereby ensuring objective and reliable assessments.
>
> Furthermore, to validate the reliability of our LLM judge, we compare the evaluation results against those from human experts on our VisualProbe benchmark. The results show a high degree of consistency, with an agreement rate of **98%**. This strong correlation confirms that using an LLM for reward scoring is a dependable and trustworthy approach for our task.

---

> ### Author Response · Authors · 2025-11-22
> **Rebuttal by Authors (Part 2 of 3)**
>
> > ### [Weakness 2] Qualitative Analysis of Reasoning Behaviors and Depth
>
> Thank you for your valuable suggestion and for acknowledging our quantitative experiments. In response, we have conducted further qualitative analysis on reasoning behaviors, focusing on **trajectory diversity**, **failure patterns**, and the **relationship between reasoning depth and success rate**. We present our findings below.
>
> **1. Trajectory diversity.**
>
> We demonstrate the diversity of our model's reasoning trajectories in **Appendix A.6 (Figs. 9, 10, and 11)**. These examples showcase our model's ability to perform **sophisticated trial-and-error exploration**. Specifically, our model dynamiclly navigates different regions of an image. When it determines that the current observation does not contain the target, it backtracks to use a previous image from the history and initiates a new zoom-in operation to explore a different area.
>
> A compelling illustration of this is Example 3 (Fig. 11), where the model engages in an 18-turn reasoning process. It meticulously examines various regions of the image, demonstrating robust and diverse reasoning behaviors before ultimately arriving at the correct answer. This highlights the model's powerful reasoning capabilities and its ability to generate rich, non-repetitive exploratory paths.
>
> **2. Failure patterns.**
>
> One failure mode we observed, though infrequent, is the emergence of repetitive loops. In such cases, the model can get stuck in a cycle of meaningless reflection and backtracking. Below is part of an example demonstrates such a reasoning loop:
>
> > **Failure Pattern Example**
> >
> > **Question:** What is the number below "67.200"?
> >
> > **Ground Truth:** 2.22.
> >
> > **Part of Model Response:**
> >
> > > The user is asking for the number below the string "67.200". In Observation 2, I can clearly see a red container with text and numbers. On the side of the container, inside a blue rectangle, I can read "FCIU 352849 7". However, this does not contain the number "67.200". I need to re-examine the previous observations to find where this number might be.
> > >
> > > **Looking back at Observation 1**, I see blue containers, and some text ...
> > >
> > > **Let's try to find the string "67.200" in Observation 1 again** but in a slightly different area ...
> > >
> > >  **Let me try to widen my crop from Observation 1 to see if I miss the target label.** Wait. Let's re-examine the whole original image ...
> > >
> > > ...
> > >
> > > Wait a second, let me **re-examine observation 1** one last time ... Wait a second, **let me go back to Observation 1 from scratch again.** ...
>
> In this example, the model **repeatedly attempts to re-examine Observation 1**, intending to zoom into a different region for further analysis. However, it fails to correctly invoke the crop tool for this action. Instead, it gets stuck in a loop, repeatedly reasoning about the same Observation 1 without making progress, ultimately leading to its failure to answer this question.
>
> Notably, the occurrence of such behavior is relatively rare. The root cause might be insufficient grounding and memory utilization capability. A future direction is to enhance the fundamental capabilities of grounding and long-context processing throught mid-training. We will incorporate the discussion of failure cases into our revised version.
>
> **3. Correlation between reasoning depth and success rate.**
>
> To qualitatively analyze the correlation between reasoning depth and success rate we would like to direct the reviewer's attention to **Appendix A.6**, **Figure 11**.
>
> This example clearly demonstrates such relationship. A greater reasoning depth provides the model with more space to perform backtracking and complex exploration. As shown in the figure, the model exhibits emergent behaviors such as **coarse-to-fine zooming**, **refocusing**, and **verification across multiple observations**. This iterative process, enabled by a deep 18-turn reasoning trajectory, ultimately allows the model to successfully identify the correct answer.

---

> ### Author Response · Authors · 2025-11-22
> **Rebuttal by Authors (Part 3 of 3)**
>
> > ### [Question 1] Impact of Turn Budget on Over-turn Masking
>
> We analyze the impact of the turn budget in detail in **Appendix A.3 (Fig. 8)**, and we summarize the key findings here.
>
> Our experiments show that the choice of the turn budget presents a clear trade-off between model performance and training efficiency. Specifically:
> - A lower budget (i.e., 6 turns) enjoys much more efficient training and faster initial convergence, but the performance plateaus early.
>
> - A higher budget (i.e., 12 turns) ultimately exhibits a superior performance ceiling, although the model converges more slowly and requires longer training.
>
> Therefore, we view the turn budget not as a sensitive hyperparameter that risks instability, but as a design choice to balance between state-of-the-art performance and training efficiency.
>
> > ### [Question 2] Analysis of Degeneration in Long-turn Trajectories
>
> We have observed that the model occasionally degenerates into repetitive loops or zooming behaviors in some long-turn trajectories.
> As you noted, we have already presented an example of such a failure pattern (repetitive loops) in **[Weakness 2] (Failure Patterns)**. To provide a more comprehensive view of these failure modes, we have included additional visualizations of such cases, including both loops and repetitive zooming, in our **revised Appendix**. We kindly invite the reviewer to refer to the Appendix for a more detailed examination of these failure patterns.
>
> > ### [Question 3] Scaling Mini-o3 with a Larger Base Model
>
> To investigate the scalability on model size, we conducted a comparative experiment by integrating Mini-o3 with **Qwen2.5-VL-32B** as its base model. It was trained using the exact same data as the 7B version. We then evaluated its performance on the three difficulty levels of the VisualProbe benchmark in the following table. It reveals the fact that larger model leads to higher performance.
>
> | Base Model | VisualProbe-Easy | VisualProbe-Medium | VisualProbe-hard |
> | :--- | :--- | :--- | :--- |
> | Mini-o3-Qwen2.5-VL-7B | 67.0 | 50.4 | 48.0 |
> | Mini-o3-Qwen2.5-VL-32B | 71.4 | 57.3 | 51.6 |
>
> > ### [Question 4] Minor Typos and Grammar
>
> We sincerely appreciate your careful reading and pointing out these typos. We have corrected these and other minor writing issues throughout the revised manuscript.

---

> > ### Comment · Reviewer_ku2Y · 2025-11-24
> > **Thanks for the response**
> >
> > The author response has addressed most of my concerns, and the additional analysis and results are compelling. I will retain my score recommending acceptance.

---

> > > ### Author Response · Authors · 2025-11-26
> > >
> > > Thanks a lot for your time and efforts. If you have any further questions, please don’t hesitate to let us know.

---

### Official Review · Reviewer_SzFD · 2025-10-31

**Soundness:** 2
**Presentation:** 2
**Contribution:** 2
**Rating:** 4
**Confidence:** 2

**Summary:**

Mini‑o3 is a multi‑turn visual search system that scales reasoning to tens of steps by combining a new Visual Probe dataset, an iterative cold‑start trajectory pipeline, and an over‑turn masking RL strategy that avoids penalizing responses hitting the turn limit. Trained with only six turns, it generalizes to longer test‑time trajectories where accuracy rises as the allowed turns increase, achieving state‑of‑the‑art results on visual search benchmarks.

**Strengths:**

- Mini-o3 achieves state-of-the-art performance on several visual search benchmarks, clearly outperforming previous open models. The improvement is from allowing the model to perform more reasoning turns. The over-turn masking method improves long-turn reasoning while keeping training efficient.
- The new Visual Probe dataset introduces difficult visual search tasks. Mini-o3 trained on Visual Probe has demonstrated some sort of trial-and-error reasoning.

**Weaknesses:**

- Missing details
    - The paper relies on an LLM judge for rewards, however the exact judging prompts/criteria are not fully specified in the main text.
    - The dataset difficulty splits (easy/medium/hard) are introduced and used in results but their criteria are only briefly mentioned. I would suggest the author to add more details on how the data split are decided.
    - Similarly, the cold‑start trajectory generation details (for example, exact prompts, hand-crafted in-context examples, acceptance filters) should be included in the main text or appendix.
- I wonder whether the authors have considered some other variants for over-turn masking
    - Mask only context-overflow instead of turn-overflow
    - Soft penalty instead of zero-mask for over-turn content
- Missing ablation of removing the DeepEyes portion or changing the ratio between VisualProbe and DeepEyes from the training to see how much the mixed-data balance matters.

**Questions:**

- Interesting that ChartQA performance dropped after the RL training compared to Qwen2.5-VL. I am curious about what the authors think might be the reason? Is it because of the data domain used for RL training?
- Curious how much improvements on general perception benchmarks, like BLINK, and CVBench. and also spatial reasoning benchmarks, like SAT.

---

> ### Author Response · Authors · 2025-11-22
> **Rebuttal by Authors (Part 1 of 4)**
>
> Dear Reviewer SzFD,
>
> Thank you for the valuable feedback. We are grateful that you recognized the **state-of-the-art performance on several visual search benchmarks** and the **effectiveness of over-turn masking method** for efficient multi-turn reasoning. We appreciate your constructive suggestions and have revised our paper accordingly. Below, we address your questions in a point-by-point manner.
>
> ---
>
> > ### [Weakness 1] Missing Details
>
> Thank you for pointing out the missing details. We apologize for the lack of clarity. We have added all the requested specifics to the appendix and below are the summary of some key information.
>
> **1. Details on LLM-as-Judge for rewards:**
>
> We use an external LLM as a judge to evaluate the model's predictions against the ground truth. Notably, we only use pure text for judging during this process. The detailed prompt template is shown below:
>
> > **Prompt Template for LLM-as-Judge.** *Question*, *Ground Truth*, and *Prediction* are dynamically replaced with the specific question, ground truth, and model prediction during evaluation.
> >
> > ---
> >
> > **SYSTEM PROMPT:**
> >
> > You are an intelligent chatbot designed for evaluating the correctness of generative outputs for question-answer pairs. Your task is to compare the predicted answer with the correct answer and determine if they match meaningfully. Here's how you can accomplish the task:
> >
> > ---
> >
> > ### INSTRUCTIONS:
> >
> > - Focus on the meaningful match between the predicted answer and the correct answer.
> > - Consider synonyms or paraphrases as valid matches.
> > - Evaluate the correctness of the prediction compared to the answer.
> >
> > ---
> >
> > **USER PROMPT:**
> >
> > I will give you a question related to an image and the following text as inputs:
> >
> > 1.  **Question Related to the Image**: *Question*
> > 2.  **Ground Truth Answer**: *Ground Truth*
> > 3.  **Model Predicted Answer**: *Prediction*
> >
> > Your task is to evaluate the model's predicted answer against the ground truth answer, based on the context provided by the question related to the image. Consider the following criteria for evaluation:
> >
> > - **Relevance**: Does the predicted answer directly address the question posed, considering the information provided by the given question?
> > - **Accuracy**: Compare the predicted answer to the ground truth answer. You need to evaluate from the following two perspectives:
> >   (1) If the ground truth answer is open-ended, consider whether the prediction accurately reflects the information given in the ground truth without introducing factual inaccuracies. If it does, the prediction should be considered correct.
> >   (2) If the ground truth answer is a definitive answer, strictly compare the model's prediction to the actual answer. Pay attention to unit conversions such as length and angle, etc. As long as the results are consistent, the model's prediction should be deemed correct.
> >
> > **Output Format**:
> > Your response should include an integer score indicating the correctness of the prediction: 1 for correct and 0 for incorrect. Note that 1 means the model's prediction strictly aligns with the ground truth, while 0 means it does not. The format should be "Score: 0 or 1".
>
> When evaluating the reward score, we perform keyword matching on the evaluation of model's response. Specifically, we first match the pattern "Score: ", and then parse the subsequent number, which is either 0 or 1, to obtain the final reward score.
>
> **2. Details on defining dataset difficulty splits:**
>
> We divide the benchmark test set into three difficulty levels based on the accuracy of model's prediction on each question. Specifically, the process is as follows:
>
> 1. First, we use the base model, Qwen2.5-VL-7B-Instruct, to perform 8 rollouts for each question. Questions with a Pass@8 > 0 are classified as the easy split. The remaining questions (where Pass@8 = 0) are retained for the next step.
>
> 2. Next, we fine-tune the base model on our VisualProbe-Train dataset via single-turn reinforcement learning. We then take a model checkpoint from the middle of this training process.
>
> 3. Finally, we use this mid-training checkpoint to perform another 8 rollouts on the previously retained set of data. In this round, questions with a Pass@8 > 0 are classified as medium, while those still having a Pass@8 = 0 are classified as hard.

---

> > ### Author Response · Authors · 2025-11-22
> > **Rebuttal by Authors (Part 2 of 4)**
> >
> > **3. Details on cold-start trajectory generation:**
> >
> > Thanks for pointing out the missing details for cold-start trajectory generation. Below we provide the exact prompt template:
> >
> > > You are an agent that answers the questions by using following two actions:
> > > 1. grounding([x1, y1, x2, y2], source): Return the bounding box [w1, h1, w2, h2] for the region on the 'source' observation image, where (w1, h1) and (w2, h2) are the top-left and bottom-right coordinates (note: w and h represent the width and height; the width and height of the image are normalized to 1.0; the coordinates of top-left corner are (0, 0).), and 'source' could be either "original_image" or "observation_i", meaning one of the history observations;
> > >
> > > 2. final(answer): return the final answer.
> > >
> > > Your generation should have the following format:
> > >
> > > Thought: to analyze the question & observation and inform the action.
> > >
> > > Action: your action
> > >
> > > ------
> > > Here are several examples:
> >
> > The hand-crafted in-context examples is organized as follows:
> >
> > > ---
> > >
> > > Example *N*:
> > >
> > > Question: *Question*
> > >
> > > Original Image: *Original Image*
> > >
> > > Thought *i*: *Thought*
> > >
> > > Action *i*: *Action*
> > >
> > > Observation *i*: *Observation*
> >
> > To ensure the diversity of examples, we selected 4 different samples to prompt the model. Finally, we provide the following prompts to structure model responses and simplify key information extraction.
> >
> > > Thought *i*:
> > >
> > > Action *i*:
> > >
> > > Please follow the language style of the above examples to finish the thought and action.
> > >
> > > You should mimic human to think carefully and comprehensively. Output the thought **as detailed as possible**, along with some useful description of the observation, suspective conjectures, and underlying world knowledge that might be useful for problem solving.
> > >
> > > If the target object is present in observation *i*, please analyze whether final answer is ready or further grounding is needed.
> > >
> > > Otherwise, you can go back to either original image or any of the history observations to find another region. In this case, clarify this thinking process, and analyze how the old bounding box deviates from the correct one, and how you obtain the coordinates of the new bounding box in Thought *i*.
> > >
> > > The output format should be in JSON as:
> > > ```json
> > > {
> > >     "thought": "xxx",
> > >     "action": "grounding([x1, y1, x2, y2], source='xxx')"
> > > }
> > > ```
> >
> > Thus, we can directly extract the reasoning content and model's next action from the JSON output.

---

> > > ### Author Response · Authors · 2025-11-22
> > > **Rebuttal by Authors (Part 3 of 4)**
> > >
> > > **3. Details on cold-start trajectory generation:**
> > >
> > > Due to space constraints, we present the reasoning output of one in-context example here. To ensure reproducibility, we have included the full text and image details for all in-context examples in the appendix.
> > >
> > > > **Example 1:**
> > > >
> > > > **Question:** What is written in English on the top first line of the right side of the wooden wall in the middle of the picture?
> > > >
> > > > **Original Image:** *Original Image*
> > > >
> > > > **Thought 0:** The user's question asks about the text on the right side of the wooden wall. To analyze it, I may need to zoom into the center part of the image. It seems like the text is too small, so I'll have to crop that region and possibly enlarge the image to read it properly. It could be something like "meeting room" or "conference room", but I'll need to visually inspect and confirm.
> > > >
> > > > I see there might be small metal letters on the wooden wall, likely on the right side. To check the text, I'll crop the image around the dimensions of 0.45 - 0.65 x 0.3 - 0.6. After cropping, I'll zoom in to make it easier to read. This should help me see the text clearly, allowing me to provide the most accurate answer based on this zoomed-in section of the image.
> > > >
> > > > **Action 0:** grounding([0.45, 0.3, 0.65, 0.6], source="original_image")
> > > >
> > > > **Observation 1:** *Observation 1*
> > > >
> > > > **Thought 1:** I see small metal letters on the right side of the central wooden wall. To focus on just the text, I'll crop the image more narrowly, focusing only on the text area. By narrowing the crop horizontally, I can zoom in specifically on the text, which should help me read it more clearly. This should allow me to zoom in and get a more detailed and accurate result. The target text is located approximately at the top half of this image, so let's focus on the range from 0.25 to 0.5 of the height dimension in the image of observation 1.
> > > >
> > > > **Action 1:** grounding([0.05, 0.25, 0.95, 0.5], source="observation_1")
> > > >
> > > > **Observation 2:** *Observation 2*
> > > >
> > > > **Thought 2:** I noticed there might be small metal letters on the right side of the central wooden wall, which is facing the camera. But I could only see the first line of English text partially in observation 2. So I should return to observation 1 and try a better bounding box. The previous bounding box is [0.05, 0.25, 0.95, 0.50], and the resulting observation seems too high, so I should move the top left corner of the bounding box downwards from (0.05, 0.25) to (0.05, 0.35), and the bottom right corner from (0.95, 0.50) to (0.95, 0.55). Therefore, the new bounding box should be [0.05, 0.35, 0.95, 0.55] to better see the text. Once cropped, I can display it and zoom in to get a clearer view of the text. This should help me identify the text in greater detail!
> > > >
> > > > Action 2: grounding([0.05, 0.35, 0.95, 0.55], source="observation_1")
> > > >
> > > > **Observation 3:** *Observation 3*
> > > >
> > > > **Thought 3:** Upon closer inspection, I can clearly see the English text on the top first line is "Meeting Room". The final answer is confirmed. No further checking is needed.
> > > >
> > > > Action 3: final("Meeting Room")
> > >
> > > By leveraging the prompt template described above to hint the model, we can exploit its in-context learning capability to **trigger calls to crop tool**. The model **exhibits similar trial-and-error behaviors** during inference. With contextual hints, we allow the model to rollout on a subset of the training data. We then **employ an LLM as a Judge** to compare the model's final output against the Ground Truth (Using the judgment prompt template introduced in *Weakness 1*), retaining only the correct results as the cold-start training data.

---

> > > > ### Author Response · Authors · 2025-11-22
> > > > **Rebuttal by Authors (Part 4 of 4)**
> > > >
> > > > > ### [Weakness 2] Other variants of Over-turn Masking
> > > >
> > > > We follow your valuable suggestions and conduct more ablations on context-overflow masking and soft penalty as shown in the following table.
> > > >
> > > > **1. Context-overflow masking**: By comparing the experiments 1, 2, and 4, we can clearly see that the context-overflow masking yields similar results as the baseline model, and lags far behind the proposed over-turn masking on all benchmarks. Actually, during RL training, since we set the upper bound of interactions to a small number (i.e., 6), the proportion of context-overflow trajectories is nearly zero, which means the training nearly degrades into the setting w/o over-turn masking.
> > > >
> > > > **2. Soft-penalty**: Following your valuable suggestion, we apply the soft penalty by multiplying the advantages of over-turn content by $0.1$ as shown in Experiment 3. By comparing experiments 1, 3, and 4, we can see that soft-penalty is useful for the accuracy of hard problems (i.e., VisualProbe-hard). However, it still falls behind the technique of over-turn masking.
> > > >
> > > > | Exp ID | Method | VisualProbe-Easy | VisualProbe-Medium | VisualProbe-hard |
> > > > | :--- | :--- | :--- | :--- | :--- |
> > > > | 1 | Baseline (w/o Over-turn Masking) | 61.1 | 45.7 | 32.2 |
> > > > | 2 | Context-overflow Masking | 60.9 | 46.0 | 33.1 |
> > > > | 3 | Soft-penalty | 61.3 | 45.0 | 41.5 |
> > > > | 4 | Over-turn Masking | 67.0 | 50.4 | 48.0 |
> > > >
> > > > > ### [Weakness 3] Ablation on DeepEyes Training Data
> > > >
> > > > Thank you for pointing out the question on data mixture. The ablation of removing the DeepEyes portion is conducted as follows. With DeepEyes-4K-Train, we observe a significant performance boost on VisualProbe-Easy and V* Bench, since DeepEyes-4K-Train mainly consists of easy tasks and relatively low-resolution images.
> > > >
> > > > | Training Data | VisualProbe-Easy | VisualProbe-Medium | VisualProbe-hard | V* Bench |
> > > > | :--- | :--- | :--- | :--- | :--- |
> > > > | VisualProbe-Train | 64.3 | **50.5** | 44.3 | 82.1 |
> > > > | + DeepEyes-4K-Train | **67.0** | 50.4 | **48.0** | **88.2** |
> > > >
> > > > ---
> > > >
> > > > > ### [Question 1] Explanation of Model Performance on ChartQA
> > > >
> > > > Thank you for pointing out the issue of data domain. We attribute slight performance drop on ChartQA to our training data proportion and data distribution. Specifically, our training data mainly covers real-world scenarios, while chart-type data is relatively scarce, which might lead to the slight performance drop on this task.
> > > >
> > > > Additionally, the primary focus of this work is not on building a strong and comprehensive model across all benchmarks, as it involves a lot of data engineering. Instead, we aim to validate the benefits of scaling up the turns and reasoning patterns, revealing the test-time scaling property on dimensions other than model size and data quantity. It is believed that with large-scale, diverse, and high-quality training data, our model would achieve excellent performance on a variety of benchmarks.
> > > >
> > > > > ### [Question 2] Model Performance on Additional General Perception Benchmarks
> > > >
> > > > Following your valuable suggestion, we evaluate the performance of Mini-o3 on CV-Bench, BLINK validation split and SAT validation split. To ensure a fair comparison, we employ an external LLM to compare the model's prediction against the ground truth answer. To reduce variance caused by randomness, we report Avg@8. The detailed results are shown in the table below:
> > > >
> > > > | Model | CV-Bench | BLINK-val | SAT-val |
> > > > | :--- | :--- | :--- | :--- |
> > > > | Qwen2.5-VL-7B-Instruct | 73.9 | **47.1** | 56.0 |
> > > > | Mini-o3 | **74.4** | 46.9 | **64.1** |
> > > >
> > > > The results demonstrate that Mini-o3 achieves better results on CV-Bench and SAT-val and similar results on BLINK-val. The pottential reason might be that the questions in BLINK-val involve images as answer options, but our model has not been exposed to such data during both SFT and RL phases.

---

### Meta-Review · Area_Chair_Lxpy · 2026-01-07

**Summary:**

This paper presents Mini-o3, a system that supports multi-turn visual reasoning to solve complex visual search tasks. This paper demonstrates that Mini-o3 produces rich, scalable reasoning patterns that achieve state-of-the-art performance through improved tool-based interactions and innovative training strategies.

The major strengths highlighted by the reviewers include comprehensive experiments, a clear and reproducible training recipe, a turn-scaling property (with over-turn masking), and the VisualProbe dataset.

Some concerns were raised during the initial review:
- generalization to other reasoning domains,
- robustness to perturbation, and
- performance of different base model sizes.

These major concerns have been adequately addressed in the rebuttal with additional experimental results.

Based on the reviews, the AC believes that the paper is worthy of acceptance to ICLR.

**Reviewer Concerns:**

- generalization to other reasoning domains,
- robustness to perturbation, and
- performance of different base model sizes.

I believe that these concerns have been resolved in the rebuttal.

**Reviewer Scores:**

Based on the summary, the AC believes that all reviewers will maintain or increase their scores.

Reviewer SzFD: rating 4
- Concern 1: Missing details
-- The rebuttal provided the requested information on the LLM-as-judgee for rewards, dataset difficulty splits, and cold-start trajectory generation.

- Concern 2: Variants for over-turn masking
-- The authors provided ablations on context-overflow masking and soft penalty

- Concern 3: Ablation of removing the DeepEyes
-- Results provided in the rebuttal.

Reviewer ku2Y: rating 6
- "The author response has addressed most of my concerns, and the additional analysis and results are compelling. I will retain my score recommending acceptance."

Reviewer iPEo: rating 8
- very positive scores, but since the reviews are excessively short, the AC was discounted in the decision process.

Reviewer XKVc
- rating 6 "I thank the authors for the responses. I have no further questions. "

---

### Decision · Program_Chairs · 2026-01-26

Accept (Poster)